

# Seawater Analysis by Ambient Mass Spectrometry-Based Seaomics and Implications on Secondary Organic Aerosol Formation

Nicolás Zabalegui[1,2], Malena Manzi[1,3], Antoine Depoorter[4], Nathalie Hayeck[4&], Marie Roveretto[4], Chunlin Li[4#], Manuela van Pinxteren[5], Hartmut Herrmann[5], Christian George[4], and María Eugenia Monge[1]

*Correspondence to:* María Eugenia Monge (maria.monge@cibion.conicet.gov.ar)

[1]Centro de Investigaciones en Bionanociencias (CIBION), Consejo Nacional de Investigaciones Científicas y Técnicas (CONICET), Godoy Cruz 2390, Ciudad de Buenos Aires, C1425FQD, Argentina.
[2]Departamento de Química Inorgánica Analítica y Química Física, Facultad de Ciencias Exactas y Naturales, Universidad de Buenos Aires, Ciudad Universitaria, Buenos Aires, C1428EGA, Argentina.
[3]Departamento de Química Biológica, Facultad de Farmacia y Bioquímica, Universidad de Buenos Aires. Junín 956, Buenos Aires, C1113AAD, Argentina
[4]Université de Lyon 1, Lyon, F-69626, France; CNRS, UMR5256, IRCELYON, Institut de Recherches sur la Catalyse et l'Environnement de Lyon, Villeurbanne, F-69626, France.
[5]Leibniz-Institut für Troposphärenforschung e.V. (TROPOS), Atmospheric Chemistry Dept. (ACD), Permoserstraße 15, Leipzig, 04318, Germany.
[&]Now at the Chemistry Department, Faculty of Arts and Sciences, American University of Beirut, Beirut, Lebanon
[#]Now at the Department of Earth and Planetary Sciences, Weizmann Institute of Science, Rehovot 76100, Israel






**Abstract.** A transmission mode-direct analysis in real time-quadrupole time of flight-mass spectrometry (TM-DART-QTOF-MS)-based analytical method coupled to multivariate statistical analysis was developed to interrogate lipophilic compounds in seawater samples without the need of desalinization. An untargeted metabolomics approach addressed here as

seaomics was successfully implemented to discriminate sea surface microlayer (SML) from underlying water (ULW) samples (n=22, 10 paired samples) collected during a field campaign at the Cape Verde islands in September-October 2017. A panel of 11 ionic species detected in all samples allowed sample class discrimination by means of supervised multivariate statistical models. Tentative identification of these species suggest that saturated fatty acids, peptides, fatty alcohols, halogenated compounds, and oxygenated boron-containing organic compounds may be involved in water-air transfer

processes and in photochemical reactions at the water-air interface of the ocean. A subset of SML samples (n=5) were subject to on-site experiments during the campaign using a lab-to-the-field approach to test their secondary organic aerosol (SOA) formation potency. Results from these experiments and the analytical seaomics strategy provide a proof of concept that organic compounds play a key role in aerosol formation processes at the water/air interface.


*Keywords:* Seaomics, Untargeted Metabolomics, Ambient Mass Spectrometry, DART, Sea Surface Microlayer, Dissolved Organic Matter, Multivariate Statistical Analysis, Secondary Organic Aerosol formation



## 1 Introduction

Oceans act as sinks and sources for gases and aerosol particles. The ocean surface chemical composition influences physicochemical processes occurring at the air-water interface by connecting the ocean biogeochemistry with the atmospheric chemistry in the marine boundary layer (MBL) (Donaldson and George, 2012). Therefore, understanding how organic compounds of marine origin are influencing the formation of secondary aerosols in the MBL with potential impacts on the radiative fluxes, aerosol hygroscopicity and subsequent cloud condensation nuclei properties is important. It has been

suggested that complex photoactive compounds are enhanced at the air-sea interface (Reeser et al., 2009a;Reeser et al., 2009b), inducing abiotic production of volatile organic compounds. For instance, experimental photosensitized reactions at the air-water interface using humic acids as a proxy of dissolved organic matter (DOM), have led to the chemical conversion of linear saturated fatty acids into unsaturated functionalized gas phase products (Ciuraru et al., 2015). Atmospheric photochemistry can even take place in the absence of photosensitizers if the air-water interface is coated with a fatty acid

(Rossignol et al., 2016). On a global scale, interfacial photochemistry has recently been proven to serve as an abiotic source of volatile organic compounds comparable to marine biological emissions (Brüggemann et al., 2018).

The sea surface microlayer (SML) covers up to 70 % of the Earth's surface and is enriched in DOM, including organic compounds derived from oceanic biota, UV-absorbing humic-like substances, fatty acids, amino acids, proteins, lipids, phenolic compounds, particulate matter, microorganisms (Liss and Duce, 2009;Donaldson and George, 2012), colloids and

phytoplankton-exuded aggregates, mainly constituted by lipopolysaccharides (Liss and Duce, 1997;Hunter and Liss, 1977;Bayliss and Bucat, 1975;Liss, 1986;Hardy, 1982;Garabetian et al., 1993;Williams et al., 1986;Schneider and Gagosian, 1985;Gershy, 1983;Guitart et al., 2004;Facchini et al., 2008;Kovac et al., 2002). While the identification of these classes of compounds has been achieved in the past, an improved chemical characterization of the SML and its chemical processing is highly desirable to understand its contribution into atmospheric composition, air quality and climate change (Liss and Duce,

70    2009).

Metabolomics is the comprehensive analysis and characterization of all small molecules (MW<1500) in a biological system, (Fiehn et al., 2000;Nicholson and Lindon, 2008) such as the marine metabolome. Mass spectrometry (MS) is one of the primary analytical techniques used to explore the metabolome, as it is highly sensitive and versatile for chemical analyses in targeted and untargeted studies (Clendinen et al., 2017;Weckwerth and Morgenthal, 2005). Targeted metabolomics focuses

on detecting and quantifying a pre-selected set of metabolites. Conversely, untargeted metabolomics attempts to cover the broadest range of detectable compounds in a biological system (Viant et al., 2019), to subsequently extract chemical patterns or class fingerprints that can allow sample classification based on metabolite panels without any a priori hypotheses. Multivariate statistical techniques compute all compound features (variables) simultaneously with the aim of reducing data dimensionality, finding underlying trends, and isolating feature/metabolite panels relevant to class discrimination (Saccenti



et al., 2014). Following compound identification, relative changes of abundances can be analyzed for biological interpretation.

Advancements in new, soft ambient ion generation techniques offer alternative MS-based applications for surface analysis with little to no sample preparation, addressing high-throughput analytical challenges in untargeted metabolomics workflows
(Monge et al., 2013;Harris et al., 2011;Clendinen et al., 2017). In particular, direct analysis in real time (DART), (Cody et al., 2005;Gross, 2014;Jones et al., 2014a;Monge and Fernández, 2014) which is a plasma-based ambient ion source, has been successfully applied in untargeted metabolomics studies in different scientific fields (Salter et al., 2011;Ifa et al., 2009;Steiner and Larson, 2009;Fernández et al., 2006;Chernetsova et al., 2010;Hajslova et al., 2011;Cajka et al., 2011;Dove et al., 2012;Jones and Fernández, 2013;Zang et al., 2017), though no studies have been reported up to date exploring oceanic
biological systems. In DART-MS, a stream of metastable atomic or molecular species generated within the discharge heated He or N2, is directed at the sample, and ions are suctioned into the mass spectrometer (Cody et al., 2005). Thermally-desorbed analytes having typically MW<1000, are ionized following atmospheric pressure chemical ionization-like pathways (Cody et al., 2005;Song et al., 2009a;Song et al., 2009b;McEwen and Larsen, 2009). An important advantage of DART compared to electrospray ionization for seawater analysis is that it is less affected by high salt levels (Kaylor et al.,
2014;Tang et al., 2004), avoiding desalinization processes that may lead to sample alteration.

In the present work, a transmission mode (TM)-DART-quadrupole time-of-flight (QTOF)-MS-based analytical method was developed to interrogate seawater DOM composition in SML and underlying water (ULW) samples collected during a field campaign at the Cape Verde islands in September-October 2017. An untargeted metabolomics approach, addressed here as seaomics, was implemented to successfully discriminate SML from ULW samples based on a selected panel of 11 ionic
species. Tentative identification of the discriminant panel provided insight into the family of compounds that may be involved in water-air transfer processes and photochemical reactions at the water-air interface of the ocean surface. In addition, secondary organic aerosol formation potency from SML interfacial photochemical products was explored during the field campaign using a lab-to-the-field approach. To our knowledge, this is the first study applying an untargeted TM-DART-QTOF-MS-based seaomics analytical strategy coupled to multivariate statistical analysis to investigate DOM
seawater composition.

## 2 Experimental

### 2.1 Chemicals

LC-MS grade acetonitrile was purchased from Fisher Chemical (NC, USA). Ultrapure water with 18.2 MΩ·cm resistivity (Thermo Scientific Barnstead Micropure UV ultrapure water system, USA) was used to prepare standard solutions.
Commercial seawater (S9883), glucose, xylose, fructose, galactosamine, mannitol, L-glycine, L-alanine, GABA (γ-





Aminobutyric acid), L-serine, L-proline, L-valine, L-threonine, L-isoleucine, L-leucine, L-asparagine, L-aspartic acid, L-glutamine, L-glutamic acid, L-methionine, L-histidine, L-phenylalanine, L-arginine, L-tryptophan, 2-amino-4,5-dimethoxybenzoic acid, 2-cyanoguanidine, flecainide acetate, lacosamide, enalapril maleate, 4-bromo-phenol, and mercaptosuccinic acid were purchased from Sigma-Aldrich (St. Louis, MO, USA). Decanoic acid, docosanoic acid, dodecanoic acid, eicosanoic acid, and octadecanoic acid were purchased from Loradan AB, Inc. (Solna, Sweden). KBr was purchased from Biopack (CABA, Argentina), and phenol was purchased from CARLO ERBA Reagents S.A (Sabadell, Spain).

## 2.2 Sample Collection at the Cape Verde Field Campaign

Sea Surface Microlayer (SML) samples were manually collected by the traditional glass plate (GP) method (van Pinxteren et al., 2012) and with an automatic catamaran using the same sampling principle as GP, named MarParCat (CAT). The MarPArCat is an autonomous catamaran for sampling the SML on rotating glass plates. Larger quantities of SML samples can be collected with this method in a shorter time. Underlying water (ULW) samples were collected from 1.0 m sea subsurface during the same time window as SML samples, using both strategies; i.e., manual sampling addressed as GP, and MarParCat (Supplementary Table S1). SML and ULW samples that were collected in the same site are addressed as paired samples (Supplementary Information, Table S1). The samples analyzed in the present study (n=22) were collected between 18/09/2017 and 10/10/2017 and stored at -20 °C until processing. Information related to sampling conditions, sample salinity, pH, and temperature, is provided in Table S1.

## 2.3 Aerosol Particle Formation Experiments at the Cape Verde Islands

A subset of collected SML seawater samples were subject to on-site experiments using a lab-to-the-field approach to test whether they were photochemically active (Ciuraru et al., 2015). Before each experiment, 100 mL SML sample was conditioned to room temperature and divided into 12 aliquots. These were centrifuged at 3500 rpm and 4 oC for 25 min to exclude colloids and aggregates (particulate matter), using a 5702R centrifuge (Eppendorf Inc.). Subsequently, 2 mL surface solution was extracted from each centrifugal vessel to isolate closer representations of SML samples considering the dilution factor inherent to the collection process, i.e., SML diluted with ULW contribution, and leading to a total sample volume of 24 mL for subsequent experiments.

Sample irradiation was conducted using a cylindrical quartz cell reactor (2 cm diameter, 10 cm length, and 30 mL volume), half filled with 14 mL of SML solution, thereby recreating an air/water interface with a maximum area of 20 cm2. Experimental details of the reactor can be found elsewhere (Ciuraru et al., 2015). This quartz reactor was surrounded with UV lamps in a ventilated box, maintaining the system at a relatively constant room temperature. The interface was irradiated





by means of 210W actinic UV irradiation peaking at 350 nm (the spectrum is displayed in Fig. S1, Supporting Text 1) that was supplied by 7 low pressure mercury UV lamps (Philips) and one extra UV pen ray (UVP, Philips).

This experimental approach allowed reproducing the air-sea exchanges under quiescent conditions and investigating particle formation potentially arising from the reaction between photochemically emitted gaseous products and OH radicals. For this purpose, the quartz cell was continuously flushed with 600 sccm purified air, entraining the air-water interfacial exchanged gaseous products to a Potential Aerosol Mass (PAM) oxidation flow reactor with 254 nm light supply, addressed as OFR254. Particle formation via OH radical photochemistry in the OFR254 was monitored using a scanning mobility particle sizer (SMPS, model 3976, TSI) and one extra ultrafine condensation particle counter (UCPC, TSI 3776, d50 > 2.5 nm). A

description of the OFR254 operation and a scheme of the experimental setup are detailed in the Supplementary Information section (Fig. S2 and S3). Blank experiments were routinely conducted using ultrapure water (18.2 MΩ·cm resistivity).

**2.4 Sample Preparation for DART-MS Analysis**

Samples were thawed at 4 °C for 5 h; neither desalination nor filtration was performed. Samples were split in 8 mL aliquots

using 15 mL conical tubes and were subsequently frozen at -20 °C until lyophilization. Quality control (QC) samples were prepared by mixing equal volumes of all samples including both collection methods before sample lyophilization (QCALL) and after metabolite extraction and reconstitution in acetonitrile (QCMIX22). Chemical standard mixtures used for analytical method development and as system suitability samples (SSS) were prepared in ultrapure water for sugars, and amino acids, in methanol/water mixtures for lipids and by combining all standards from the three families of compounds (Supplementary

Information, Table S2). The sample preparation blank was prepared with ultrapure water as follows: fresh ultrapure water was stored for 2 days at -20 °C in a new plastic bottle equivalent to those used for sample collection; subsequently thawed, split in 8 mL aliquots and stored in 15 mL conical tubes at -20 °C until lyophilization. This protocol was also implemented to prepare commercial seawater samples (CSW) that were used for analytical method development. Blanks, QCs, SSS, and samples were lyophilized at 0.280 mBar during 48 h using a Christ Alpha 1-4 Freeze dryer. SML samples, ULW samples,

QCs, and SSS were lyophilized with sample blanks in different batches to evaluate possible cross-contamination. Lyophilized samples were shipped from TROPOS (Germany) to CIBION-CONICET (Argentina), where they were stored at -80 °C until TM-DART-QTOF-MS analysis. Metabolite extraction was performed using acetonitrile, and yielding a concentration factor of 6.67. Reconstituted samples were vortex-mixed during 5 min to guarantee efficient extraction, and centrifuged during 10 min at 4861 × g and 20 °C. For each sample, 500 μL of supernatant were collected for further analysis.




## 2.5 DART-MS Analysis

A DART® SVP ionization source (IonSense Inc., MA, USA) was coupled to a Xevo G2S QTOF mass spectrometer (Waters Corporation, Manchester, UK) by means of a VAPUR® interface flange (IonSense Inc., MA, USA). The DART source was operated with He as the discharge gas heated at 300 °C, and data were acquired in negative ionization mode. A transmission

mode (TM)-DART geometry was implemented for sample analysis, setting a distance of 2.5 cm in the rail holding the source. This allowed using the minimum possible DART-to-sample distance to provide the greatest sensitivity (Zang et al., 2017;Jones and Fernández, 2013). Samples were deposited in a stainless-steel mesh that was subsequently placed in a linear rail-based sampler, which was digitally controlled to minimize variance in sample position. Fig. S4 illustrates the experimental design for depositing samples in different spots of the mesh to avoid cross-contamination. A protocol for

calibrating the mass spectrometer across the range of m/z 50-850 using the DART source operated in TM was developed using a mixture of standards prepared in a water-methanol solution (1:1 v/v) that would provide almost equidistant m/z peaks. Signals of different adduct ions from 2-cyanoguanidine, enalapril maleate, mercaptosuccinic acid, 2-amino-4,5-dimethoxybenzoic acid, flecainide acetate, and lacosamide were used for the TOF calibration (Supplementary Information, Table S3). Drift correction was performed after data acquisition using stearic acid present as an ambient contaminant. The

[M-H]- adduct ion with m/z 283.2643 was chosen as a lock mass to have a high degree of accuracy in exact mass measurement. Data were acquired in continuum mode in the range of m/z 50-850, and the scan time was set to 1 s. A standard solution of enalapril 3.7 μM was used as an additional SSS and added to each mesh in spot #3 (Fig. S4) to evaluate mass accuracy of the [M-H]- ion at m/z 375.1925. The resolving power and mass accuracy of the TM-DART-QTOF-MS system were 23000 fwhm and 0.2 mDa at m/z 375.1925, respectively. Twelve spots per mesh were utilized for analysis.

Each spot contained 3 droplets of 20 μL of sample, which were dried at room temperature before analysis. The mesh holder was moved at a speed of 0.2 mm s-1 for data acquisition. Mesh #1-11 included a solvent (SV) blank (acetonitrile); a commercial seawater control; a sample preparation blank (using ultrapure water); a QCMIX22 (pooled QC sample from all reconstituted samples: 10 SML + 12 ULW), and technical triplicates of all samples (Fig. S4). As indicated in Fig. S4, mesh #12 included QCALL samples (pooled QC sample from all samples before lyophilization: 10 SML + 12 ULW samples). For

TM-DART-QTOF-MS/MS experiments, the product ion mass spectra were acquired with collision cell voltages between 10 and 40 V, depending on the analyte. Ultra-high-purity argon (≥99.999 %) was used as the collision gas. Data acquisition and processing were carried out using MassLynx version 4.1 (Waters Corp., Milford, MA, USA). Data were acquired for each spot, and acquisition over each mesh was automatically performed through synchronization between the DART software (IonSense Inc.) and MassLynx (Waters Corp.). System suitability procedures were performed to verify that the method and

associated instrumentation were fully functioning before and during the analysis of experimental samples.



## 2.6 Seaomics Data Analysis

Progenesis Bridge (Waters Corp., Milford, MA, USA) was used for data pre-processing. This software allowed defining the lock mass for drift correction after acquisition; and merged the original data into a Gaussian profile. Spectral features (m/z values) were further extracted from TM-DART-QTOF-MS data using Progenesis QI version 2.1 (Nonlinear Dynamics, Waters Corp., Milford, MA, USA). An absolute ion intensity filter was applied in the peak picking process for integration, defining a threshold for the aggregate run. Only SML and ULW samples were considered for peak picking. This process yielded 889 features (m/z) detected within samples. Subsequently, six features were removed due to high mass defect (potential salts clusters). For correction of inter-mesh effects, a quality control sample-based robust locally estimated scatterplot smoothing (LOESS) signal correction method (Dunn et al., 2011) was applied using QCMIX22 samples. This strategy allowed correcting for temporal signal fluctuation of each feature along the total acquisition time. Subsequently, features with relative standard deviation (RSD) >30 % in QCMIX22 were discarded, and only those with 5-fold average intensity in samples compared to blanks (i.e., sample preparation blanks and solvent blanks) were retained. Manual curation of features was also performed to eliminate redundancy (isotopic peaks from the same feature), to retain signals with a detected isotopic pattern, and to account for resolution limitations in the peak picking process. Moreover, only those monoisotopic peaks with intensity >103 in the continuum spectra were retained. The final curated matrix comprised of 51 features (m/z values) was normalized by total ion area. Abundance values from technical triplicates were averaged, except for SML GP2 sample, for which only two replicates were considered. The matrices obtained before and after averaging technical replicates (Data Set S1 in the Supplementary Information) were utilized to build unsupervised and supervised multivariate statistical analysis models using MATLAB R2015a (The MathWorks, Natick, MA, USA) with the PLS Toolbox version 8.1 (Eigenvector Research, Inc., Manson, WA, USA). Principal component analysis (PCA) (Richard A. Johnson) and t-distributed stochastic neighbor embedding (t-SNE) (Van Der Maaten and Hinton, 2008) techniques were used to track data quality, reduce the data dimensionality, identify potential outliers in the dataset, as well as to identify sample clusters and evaluate the analytical method reproducibility. Orthogonal projection to latent structures-discriminant analysis (oPLS-DA) (Trygg et al., 2007;Bylesjö et al., 2006;Trygg and Wold, 2002;Shrestha and Vertes, 2010) coupled with a genetic algorithm (GA) variable selection method was applied to find a feature panel that maximized classification accuracy for the binary comparison of SML and ULW samples. The selected group of discriminant features had the lowest root-mean-square error of cross-validation (RMSECV) at the conclusion of the GA variable selection process. This process was performed five different times and the selected panel yielded the lowest RMSECV and exhibited largest feature overlap with the other four panels. The parameters for GA were as follows: population size: 64, variable window width: 1, % initial terms (variables): 15, target minimum # of variables: 5, target maximum # of variables: 15, penalty slope: 0.03, maximum generations: 100, % at convergence: 50, mutation rate: 0.005, crossover: double, regression choice: PLS, # of latent variables: 5, cross-validation: contiguous, # of splits: 10, # of iterations: 10, replicate runs: 10. The oPLS-DA model was cross-validated using venetian blinds with 4 data splits, and 1 sample per blind to account for overfitting. Data were preprocessed by autoscaling prior to





PCA or oPLS-DA analysis. PCA was also performed to inspect data before and after GA variable selection (i.e., on the curated spectral feature matrix and on the discriminant feature panel). Fold changes were calculated for paired samples for each discriminant feature by comparing sample replicate average values for SML and ULW samples. Wilcoxon Paired Signed Rank Test was used to compare SML with ULW samples ($p < 0.05$). Median fold changes were calculated for each discriminant feature (Supplementary Information, Table S4).


### 2.7 Metabolite Identification Procedure

Metabolite identification was attempted for the discriminant features resulting from the GA variable selection process. Elemental formulae were generated based on accurate masses and isotopic patterns, taking into account stringent conditions for isotope ratios. For those cases in which there was overlap between isotopic peaks of different features, the isotopic

pattern was not considered for molecular formula generation. In addition, fragmentation patterns obtained from TM-DART-QTOF-MS/MS experiments were used for tentative identification.

### 3 Results and Discussion

### 3.1 TM-DART-QTOF-MS-based Method Optimization

Figure 1 illustrates the untargeted TM-DART-QTOF-MS seaomics analytical workflow implemented for the analysis of

seawater samples collected during the Cape Verde field campaign. A TM geometry was implemented to analyze samples in a flow-through fashion to increase reproducibility with lower risk of cross contamination (Zhou et al., 2010a;Zhou et al., 2010b;Jones and Fernández, 2013;Perez et al., 2010;Zang et al., 2017;Jones et al., 2014b). The analytical method development involved optimization of the ion source stabilization time, accomplished in 60 seconds; the synchronization between data acquisition and the linear rail control; the selection of He over N2 to generate the plasma, based on higher

sensitivity obtained with the former; the optimization of He temperature set at 300 °C; the selection of acetonitrile for metabolite extraction; the optimization of the solvent volume required for extraction to allow i) maximum metabolite concentration considering that the seawater metabolome is comprised of organic compounds with a wide range of physicochemical properties and levels; and ii) enough sample volume for technical replicates, QCs and tandem MS analysis; and the sample volume deposited on the mesh to maximize signal-to-noise ratio (number of sample droplets and droplet

volume). The selected OM extraction method with acetonitrile as extracting solvent favored the analysis of lipophilic compounds. In addition, to enhance the detection of organic acids, the analytical method was optimized operating the DART ion source in negative ionization mode.



## 3.2 Seawater Sample Fingerprinting

The curated data matrix, comprised of 51 features i.e., m/z values, and all sample replicates (Data Set S1 in the Supplementary Information), was used to build a PCA model that accumulated 62.29 % of the total variance in the first two principal components (PCs) (Fig. 2). The 2D scores plot illustrated in Fig. 2A shows distinguishable separation between acetonitrile blanks, sample preparation blanks, commercial seawater samples and seawater samples collected during the field campaign. Since the maximum data variance in a PCA model is in the direction of the base of eigenvectors of the covariance

matrix, the largest differences are given by seawater samples compared to blanks. However, seawater samples from the Cape Verde islands were discriminated from commercial seawater samples. In addition, QCMIX22 replicates clustered together, indicating reproducibility in the sample preparation method, high data quality, and adequate performance of the analytical platform. Moreover, overlapping of both type of QC samples (QCMIX22 and QCALL) suggested reproducibility in the sample extraction protocol. Solvent blanks from different mesh and different positions (spots) were clustered together,

suggesting negligible cross-contamination in the analysis. Results provided by the t-SNE model (Fig. 2B), which is a nonlinear dimensionality reduction technique, were in agreement with those provided by the linear transformation-based technique of PCA and emphasize the reproducibility of the developed analytical method for seawater sample analysis. This was further evidenced by the visualization of sample replicate clusters in a t-SNE model that only included SML and ULW samples (Fig. S5).

To investigate the possibility of seawater sample clustering, a PCA model was built with the 51 extracted and curated features for averaged technical replicates of SML and ULW samples. Fig. 3A shows the PCA scores plot including the first three principal components that accounted for 43.93 %, 25.08 %, and 8.40 % variance, respectively. No outliers were detected by this analysis and no sample clustering was visualized in the score plot. Thus, sample discrimination was further attempted by means of oPLS-DA coupled to a GA variable selection method to find a reduced set of features that would

allow sample classification and class membership prediction. A panel of 11 features with the lowest RMSECV was selected through the GA process. Fig. 3B shows the cross-validated prediction plot using the selected feature panel by means of a model that consisted of 5 latent variables that interpreted 82.19 % and 95.41 % variance from the X- (feature abundances) and Y- (class membership) blocks, respectively. This oPLS-DA model resulted in 100 % cross-validated accuracy, sensitivity, and specificity; therefore, there was no sample misclassification. Sample classification was further evaluated by

means of a non-supervised method using the 11 discriminant features to discard possible overfitting by the supervised multivariate model. Fig. 3C shows certain degree of sample separation into clusters in the PC3 dimension according to the seawater sample collection depth, i.e., SML or ULW.



### 3.3 SOA formation potency from SML samples

A subset of SML samples (CAT 8, GP 10, CAT 6, CAT 3 and CAT 4) that were analyzed by the TM-DART-QTOF-MS seaomics strategy were also subject to on-site experiments during the field campaign using a lab-to-the-field approach to test their SOA formation potency. The outcome of a typical SML irradiation experiment is illustrated for sample CAT 8 in Fig. 4. Different time periods (P) during which experimental parameters were modified along the experiment are indicated in the figure. In the absence of light (before P1), no particle formation was detected downstream of the preconditioned OFR (5.0

ppmv initial O3 and half-power UV light supply). However, when SML samples were exposed to actinic irradiation (periods P1-P4), particle formation was detected in the OFR254. Moreover, the particle number concentration exhibited OH exposure (OHexp)-dependent trends (P2-P3). Gaseous products were probably generated from photosensitized reactions at the SML interface, and subsequently reacted with OH radicals in the OFR254, leading to particle formation.

Because of the difficulty associated to on-site measuring total OHR (OH radical reactivity) from the cell reactor or tracing

OHexp in the OFR, we only qualitatively tested the particle generation rates with respect to various oxidation degrees by changing the UV light intensity or O3 concentration in the OFR. Assuming that photochemistry occurring at the SML interface was at steady-state, air-water exchanged gaseous products were constantly entrained into the OFR, and the estimated particle generation rates/OHexp for each period followed the trend: P1<P4<P2<P3. During P1, particle concentration gradually increased with SML illumination, and final number concentration exceeded $8 \times 10^3$ cm-3. These

particles exhibited a median diameter of several nanometers at the edge of the lower 10 nm size limit of the SMPS detection system, thus measuring the particle size distribution was not possible. During P2, UV light intensity was doubled in the OFR by turning all lamps on. A particle burst was detected with the UCPC, with a shift towards larger particle sizes. The oxidation capacity in the OFR was further enhanced by supplying additional external O3 (initial mixing ratio of 7.0 ppmv). Total particle concentration decreased while larger particles were formed. During P4, one UV lamp in the OFR was turned

off, and a sharp decrease in particle concentration was observed, but the final concentration was still higher than during P1 (Fig. 4). Particle formation was observed for CAT 8 and GP 10 SML samples. Results from the atmospheric simulation experiments conducted on SML samples were in agreement with previous laboratory studies that demonstrated air-sea interfacial driven chemistry as a source of marine secondary aerosol (Roveretto et al., 2019;Ciuraru et al., 2015;Fu et al., 2015).


### 3.4 Discriminant Compound Identification & Role in Aerosol Particle Formation

Compound identification was attempted for the 11 features of the discriminant panel. The coupling of the DART source to a high-resolution mass spectrometer allowed generating elemental formulae for unknown compounds, which together with tandem MS capability contributed to their identification. Figure S6 shows the high resolution continuum mass spectra





obtained for each of the discriminant features detected in all samples and obtained from the GA selection process. The
       analysis of fragment ions detected in tandem MS experiments together with neutral loss analysis provided information
       regarding functional groups and contributed to filter molecular formulae obtained by accurate mass, and isotopic pattern
       analysis. Table 1 describes the ionic species associated to the discriminant features and their corresponding molecular
       formulae, and provides information of product ions, and neutral and/or radical losses identified in TM-DART-QTOF-

MS/MS experiments. In addition, the table includes the family of compounds identified with a certain confidence level. In
       general, discriminant features comprised saturated fatty acids, fatty alcohols, peptides, brominated compounds and boron-
       containing organic compounds.

       An expected limitation of TM-DART-QTOF-MS analysis was associated to spectral overlap; thus, in some cases the isotopic
       pattern was not considered for compound identification. However, two different quadrupole mass windows of 6 and 1 Da

were used in tandem MS experiments to mitigate this problem. The mass window of 6 Da allowed investigating the complete
       isotopic profile with high sensitivity at expense of lower selectivity than the narrower mass window. In contrast, the mass
       window of 1 Da provided more confidence in the identification of product ions with higher selectivity at the expense of
       lower sensitivity than the broader mass window. In cases of low precursor ion intensity or quadrupole co-selection, MS/MS
       spectra were not collected (Table 1).

Different types of species were generated for desorbed and ionized analytes (M) by the plasma-based source operated in
       negative mode, including [M-H]-, [M]- and    [M]-• ionic species. The generation of a radical anion, [M]-•, was suggested
       for feature # 4 based on the product ions detected in tandem MS experiments and the generated molecular formulae. Based
       on the tentative identification of feature # 4, additional experiments were performed with chemical standards including a
       dicarboxylic acid (succinic acid) and saturated fatty acids under the same experimental conditions as for seawater sample

analysis. Different ionic species were detected in these experiments except for radical anions. However, literature evidence
       suggests the production of radical anions based on electron capture mechanisms occurring in He-based plasma sources
       (Cody and Dane, 2016;Bridoux and Machuron-Mandard, 2013;Jorabchi et al., 2013).

       Based on the analysis of the isotopic patterns and tandem MS results, several features were identified as oxygenated boron-
       containing organic compounds. In these compounds, the boron atom is speculated to be functionalized with saturated fatty

acids yielding tetra coordinated boron esters that would generate [M]- anions. Boron-containing compounds are known to be
       ubiquitous in vascular plants, marine algal species, and microorganisms (Dembitsky et al., 2002). Four out of five features
       identified as boron-containing organic compounds functionalized with saturated fatty acids as well as features identified as
       fatty alcohols were enriched in SML samples compared to ULW samples (Table S4).

       Compounds having a bromine atom in their molecular formula were also tentatively identified in the discriminant panel and

are suggested to be halogenated compounds rather than bromine adduct ions. This hypothesis is based on the results yielded
       by the comparative analysis of a saturated acetonitrile solution with KBr and 2 mM phenol and the analysis of an acetonitrile
       solution of 4-bromo-phenol (Fig. S7), used as model compounds. The [M-H]- ion was detected in the analysis of 4-bromo-
       phenol, but the [M+Br]- adduct ion was not observed for the KBr saturated solution containing phenol. The two features (#





21 and 34) that were identified as halogenated compounds were enriched in SML samples (Table S4). Possible sources of

halogenated compounds in SML samples are photochemical reactions occurring at the water/air interface (Roveretto et al., 2019;Donaldson and George, 2012). It is worth noting that organic compounds identified in the discriminant panel may have derived both from the secreted (exometabolome) and/or intracellular metabolites (endometabolome) of biological organisms such as algal species and microorganisms present in seawater.

Putative identification of the discriminant panel capable of differentiating SML from ULW samples provides further

evidence to support secondary organic aerosol (SOA) formation detected with the lab-to-the-field approach during the campaign. The PCA scores plot illustrated in Fig. 5 shows that SML samples were not distinguished based on the collection method, i.e., GP or CAT, and points out those SML samples that were also evaluated for SOA formation during the field campaign. As previously discussed, two of these SML samples (CAT8 and GP10) yielded SOA formation (Fig. 4). Further analysis on samples analyzed by both TM-DART-QTOF-MS and the lab-to-the-field approach suggest differences in

compound concentration levels between SML samples that led to SOA formation from those that did not (Fig. S8, Table S5). Figure S8A shows that PC2 clearly separates samples according to SOA formation. Those features that mainly contribute to sample class separation with largest absolute values in the loadings plot associated to PC2, and illustrated in Fig. S8B, were putatively identified as boron-containing organic compounds (Table S5). Despite the limitations associated with the low number of samples used to perform statistical analysis, results suggest that SML samples that led to particle formation were

enriched on boron-containing organic compounds and other unidentified molecules (Table S5). These results provide a proof of concept that organic compounds play a key role in aerosol formation process at the water/air interface.

## 4 Conclusions

An untargeted TM-DART-QTOF-MS-based analytical method coupled to multivariate statistical analysis allowed analyzing organic compounds present in SML and ULW seawater samples collected during a field campaign at the Cape Verde islands,

without the need of desalinization. This seaomics approach was successfully implemented to discriminate SML from ULW samples. Tentative identification of the discriminant metabolite panel suggests that halogenated compounds, fatty alcohols, and oxygenated boron-containing organic compounds may be involved in water-air transfer processes and in photochemical reactions at the water-air interface of the ocean. Combined results from TM-DART-QTOF-MS and on-site SOA formation testing experiments on SML samples, suggest that organic compounds enriched at the water/air interface play a key role in

aerosol formation process. This strategy, implemented for the first time in this collaborative study, provides new opportunities for improving the characterization of seawater OM content, and discovering compounds involved in aerosol formation processes.

**Data availability**

The mass spectrometry data have been deposited to the MetaboLights public repository, with the data set identifier
MTBLS1198.

**Author contribution**

MEM, MVP, HH, and CG designed the collaborative study. MVP and HH designed sample collection methods. MM
processed samples until storage at -80 °C. MEM, MM and NZ developed the TM-DART-MS-based seaomics strategy and
analyzed the data. MEM, NZ, MM, AD, NH and CG contributed to optimize the TM-DART-MS-based analytical method.
NZ and MM conducted TM-DART-MS and MS/MS experiments. MR, CL and CG conducted on-site aerosol particle
formation experiments. MEM, NZ and CG wrote the manuscript. All authors revised the manuscript.

**Competing interests**

The authors declare that they have no conflict of interest.

**Acknowledgements**

M.E.M. is a research staff member from CONICET (Consejo Nacional de Investigaciones Científicas y Técnicas,
Argentina). Funding is acknowledged to the Marie Skłodowska-Curie action of the program Research and Innovation Staff
Exchange (RISE), Horizon 2020, H2020-MSCA-RISE-2015, which financed the European network entitled "MARSU",
acronym of MARine atmospheric Science Unravelled: Analytical and mass spectrometric techniques development and
application. This project has received funding from the European Union's Horizon 2020 research and innovation
programme under grant agreement N°690958. Funding from the National Mass Spectrometry System (SNEM), CONICET
and MINCyT (project E-AC12), as well as from the Leibniz Association SAW project MarParCloud (SAW-2016-TROPOS-
2) is gratefully acknowledged. Nadja Triesch and Sebastian Zeppenfeld of TROPOS ACD are acknowledged for their
support during SML collection and sample preparation. Coretta Bauer Helmholtz Centre for Environmental Research – UFZ
is acknowledged for assisting with sample lyophilization.

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

**Table 1:** Identification of discriminant features based on accurate mass (a), isotopic pattern (b), and MS/MS experiments (c). Features with p <0.05 are highlighted in red (Wilcoxon Paired Signed Rank Test). Fold change trends in binary comparisons are indicated with arrows, ( ↑ ): increased levels, and ( ↓ ): decreased levels.

| ID[1] | $m/z$[2] | Ion type | MS — Experimental $m/z$ | Tandem MS Experiments — $m/z$ of Ions detected in MS/MS spectra (MF[3] of fragment ion; $\Delta m$/mDa) | Mass Loss[4] (MF; $\Delta m$/mDa) | Quadrupole mass window | Tentative MF for Detected Ionic Species; $\Delta m$/mDa | Criteria to obtain MF: a) exact mass b) isotopic pattern c) MS/MS information | Tentative ID: Main Class (Sub Class) | Fold Change Trend |
|---|---|---|---|---|---|---|---|---|---|---|
| 4 | 258.1835 | $[M]^-$ / $[M-H]^-$ | 258.1794 | 258.1794 ($C_{14}H_{26}O_4$; -6.5) | 16.9991 (OH; -3.6) 43.9925 ($CO_2$; -2.7) | ~ 6 Da & 1 Da | 1:$C_{14}H_{26}O_4$; -6.5 2:$C_{12}H_{24}N_3O_3$; 1.7 | a, c | 1:Fatty Acids and Conjugates (Dicarboxylic Acid) 2:Dipeptide | ↓ |
| | | | | 241,1805 ($C_{14}H_{25}O_3$; 0.1) | 43.9890 ($CO_2$; -0.8) | | | | | |
| | | | | 225.1861 ($C_{14}H_{25}O_2$; 0.6) | 42.0467 ($C_3H_6$; -0.3) | | | | | |
| | | | | 214.1868 ($C_{13}H_{26}O_2$; -6.5) | | | | | | |
| | | | | 197.1915 ($C_{13}H_{25}O$; 1.0) | | | | | | |
| | | | | 183.1402 ($C_{11}H_{19}O_2$; 1.7) | | | | | | |
| | | | | 169.1218 ($C_{10}H_{17}O_2$; -1.1) 155.1105 ($C_9H_{15}O_2$; 3.3) 141.0933 ($C_8H_{13}O_2$; 1.7) 127.0738 ($C_7H_{11}O_2$; -2.1) 113.0614 ($C_6H_9O_2$; 1.1) 99.0463 ($C_{10}H_{17}O_2$; 1.7) | 14.0109 - 14.0195 ($CH_2$, -4.8 - 3.8) | | | | | |
| 5 | 275.1652 | $[M-H]^-$ | 275.1640 | 275.1652 ($C_{17}H_{23}O_3$; 0.5) | 43.9899 ($CO_2$; 0.1) | ~ 6 Da & 1 Da | $C_{17}H_{23}O_3$; 0.5 | a, b, c | Fatty Alcohols | ↑ |
| | | | | 231.1741 ($C_{16}H_{23}O$; -0.8) | 56.0633 ($C_4H_8$; 0.7) | | | | | |
| | | | | 175.1153 | | | | | | |
| **17** | 455.4106 | $[M-H]^-$ | 455.4122 | Co-selection in quadrupole (1 Da mass window) limits interpretation | - | - | 1:$C_{28}H_{55}O_4$; 2.2 2:$C_{24}H_{51}N_6O_2$; 3.3 | a, b | NO ID | ↑ |
| 21 | 512.2138 | $[M]^-$ / $[M-H]^-$ | 512.2135 | 283.2629 ($C_{18}H_{35}O_2$; -0.8) | | ~ 6 Da & 1 Da | $C_xH_yO_zBN_wF_dP_jS_k$ $Cl_rBr$ x:[31-12];y:[34-47];z:[0-7];w:[1-9];d:[0-6];j:[0-3];k:[0-2];r:[0-1]; 2.6 | a, b, c | Brominated compound | ↑ |
| | | | | 255.2317 ($C_{16}H_{31}O_2$; -0.7) | | | | | | |
| | | | | 227.2007 ($C_{14}H_{27}O_2$; -0.4) | | | | | | |
| | | | | 78,9186 (Br, 0.4) | | | | | | |
| **25** | 557.4069 | $[M-H]^-$ | 557.4033 | Co-selection in quadrupole (1 Da mass window) limits interpretation | - | - | $C_{31}H_{57}O_8$; 1.6 | a, b | NO ID | ↓ |
| 28 | 639.4312 | $[M]^-$ | 639.4299 | 400.2036/401.2002 ($C_{22}H_{30}BO_4S$; 4.4) | | ~ 6 Da & 1 Da | $C_{38}H_{60}BO_5S$; 4.4 | a, b, c | Boron-containing organic compound | ↑ |
| | | | | 283.2672 ($C_{18}H_{35}O_2$; 3.5) | | | | | | |
| | | | | 281.2526 ($C_{18}H_{33}O_2$; 4.5) | | | | | | |
| | | | | 255.2368 ($C_{16}H_{31}O_2$; 4.4) | | | | | | |





| | | | | | | | | | | |
|---|---|---|---|---|---|---|---|---|---|---|
| | | | 227.2048 ($C_{14}H_{27}O_2$; 3.7) | | | | | | | |
| **31** | 653.5088 | [M]$^-$ | 653.5186 | 428.3046/429.3061 | | ~ 6 Da & 1 Da | $C_{37}H_{70}BO_8$; 2.2 | a, b, c | Boron-containing organic compound | ↑ |
| | | | | 382.2994/383.2968 ($C_{21}H_{40}BO_5$; -0.1) | 270.2196 ($C_{16}H_{30}O_3$; 0.1) | | | | | |
| | | | | 368.2353 | | | | | | |
| | | | | 354.2714/355.2659 ($C_{19}H_{36}BO_5$; 0.3) | 298.2505 ($C_{18}H_{34}O_3$; -0.3) | | | | | |
| | | | | 326.2000/327.1988 | | | | | | |
| | | | | 297.2426 ($C_{18}H_{33}O_3$; -0.4) | | | | | | |
| | | | | 283.2637 ($C_{18}H_{35}O_2$; 0.0) | | | | | | |
| | | | | 269.2120 ($C_{16}H_{29}O_3$; 0.3) | | | | | | |
| | | | | 255.2327 ($C_{16}H_{31}O_2$; 0.3) | | | | | | |
| | | | | 241.1799 ($C_{14}H_{25}O_3$; -0.5) | | | | | | |
| | | | | 227.2015 ($C_{14}H_{27}O_2$; 0.6) | | | | | | |
| | | | | 116.0399/117.0365 ($C_3H_6BO_4$; 0.7) | | | | | | |
| | | | | 71.0142 ($C_3H_3O_2$; 1.6) | | | | | | |
| **33** | 667.5346 | [M]$^-$ | 667.5333 | 400.2886/401.2694 ($C_{20}H_{38}BO_7$; -1.7) | | ~ 6 Da & 1 Da | $C_{38}H_{72}BO_8$; 1.9 | a, b, c | Boron-containing organic compound | ↑ |
| | | | | 428.3062/429.3033 ($C_{22}H_{42}BO_7$; 0.9) | | | | | | |
| | | | | 410.2963/411.2922 ($C_{22}H_{40}BO_6$; 0.4) | 256.2422 ($C_{16}H_{32}O_2$; 2.0) | | | | | |
| | | | | 400.2017/401.1998 (interference) | | | | | | |
| | | | | 370.2647/371.2611 ($C_{19}H_{36}BO_6$; 0.6) | | | | | | |
| | | | | 283.2643 ($C_{18}H_{35}O_2$; 0.6) | | | | | | |
| | | | | 255.2328 ($C_{16}H_{31}O_2$; 0.7) | | | | | | |
| | | | | 227.2014 ($C_{14}H_{27}O_2$; 0.4) | | | | | | |
| | | | | 116.0400/117.0365 ($C_3H_6BO_4$; 0.6) | | | | | | |
| | | | | 117.0134 ($C_4H_5O_4$; -5.4) | | | | | | |
| | | | | 78.9189/80.9168 (Br; 0.7) | | | | | | |
| | | | | 75.0083 ($C_2H_3O_3$; -0.2) | | | | | | |
| | | | | 71.0138 ($C_3H_3O_2$; 0.6) | | | | | | |
| 34 | 675.4587 | [M-H]$^-$ | 675.4571 | 315.2525 | | ~ 6 Da & 1 Da | $C_{41}H_{72}SBr$; 3.3 | a, b, c | Brominated compound | ↑ |
| | | | | 283.2677 ($C_{18}H_{35}O_2$; 0.4) | | - | | | | |
| | | | | 78.9205 (Br; 0.3) | | | | | | |
| 43 | 751.6276 | [M]$^-$ | 751.6260 | Co-selection in quadrupole (1 Da mass window) limits interpretation | | - | $C_{44}H_{84}BO_8$; 1.8 | a, b | Boron-containing organic compound | ↓ |
| **49** | 795.7092 | [M-H]$^-$ | 795.7068 | Co-selection in quadrupole (1 Da mass window) limits interpretation | | - | $C_{49}H_{95}O_7$; 0.4 | a, b | NO ID | ↓ |

[1]Feature code, [2]*m/z* value obtained from Progenesis QI, [3]Molecular Formula, [4]Possible neutral losses are indicated in parentheses



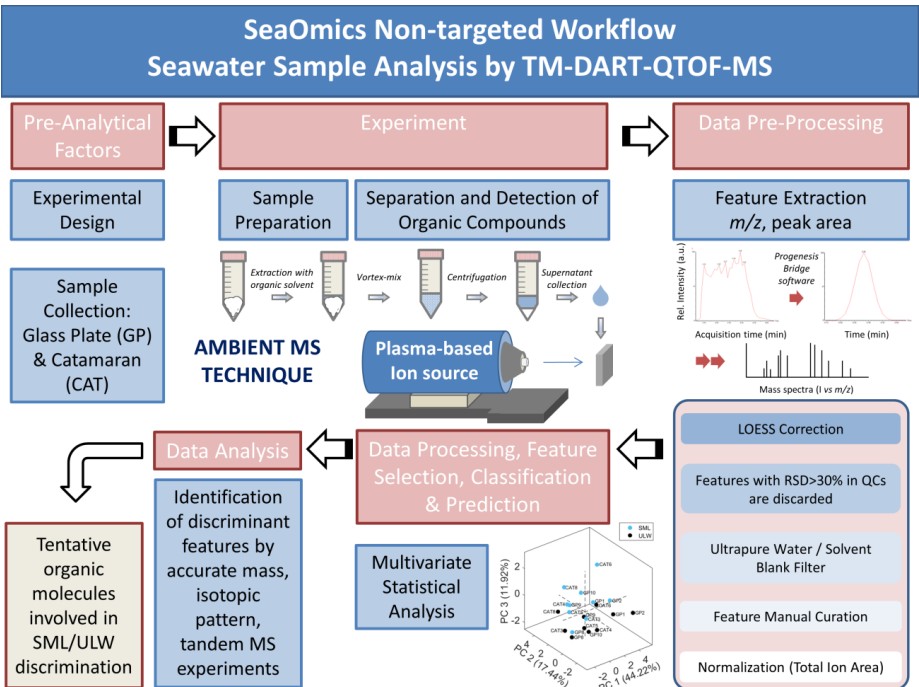

**Figure 1.** Scheme illustrating the analytical strategy implemented at CIBION-CONICET for the analysis of seawater samples using TM-DART-QTOF-MS.





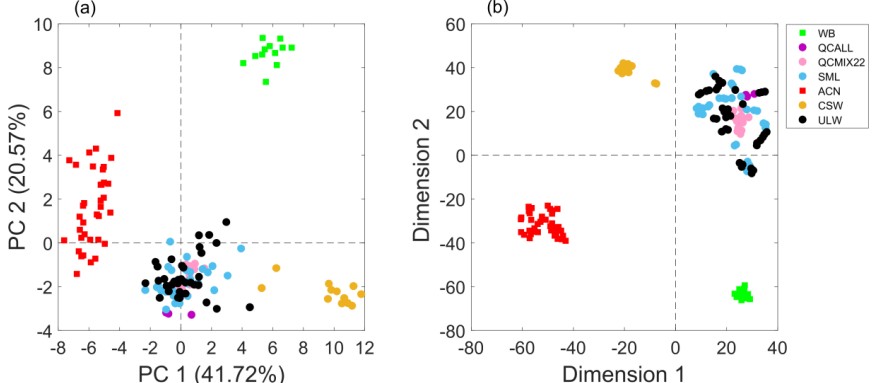

**Figure 2.** (a) PCA scores plot showing the first two principal components, and (b) bi-dimensional t-SNE plot of seawater samples (circles) and solvent blanks (squares). WB: sample preparation blanks using ultrapure water (green), QCALL: Pooled sample from all seawater samples before lyophilization (purple), QCMIX22: Pooled sample from all reconstituted seawater samples (pink), SML: Sea surface microlayer water samples (light blue), ACN: Acetonitrile (orange), CSW: Commercial seawater samples (gold), ULW: Underlying water samples (black). PCA and t-SNE models were built using the 51 extracted features and all replicates were included.





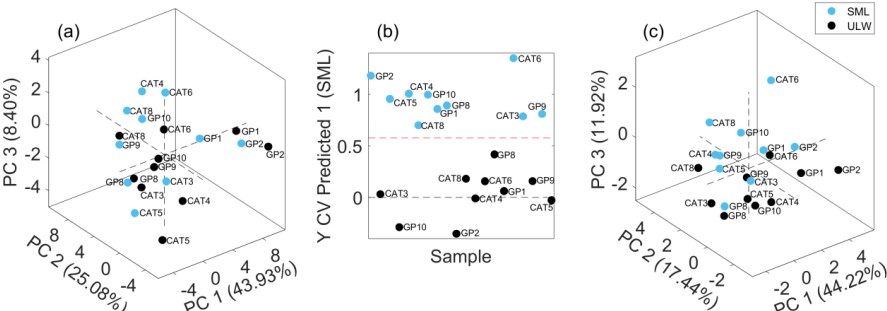

**Figure 3.** (a) PCA scores plot showing the first three principal components of sea surface microlayer samples (SML, light blue) and ultralow seawater samples (ULW, black). PCA was done based on 51 extracted features with averaged values from technical replicates. Accounted variance: PC 1, 43.93 %; PC 2, 25.08 %; PC 3, 8.40 %. (b) Cross-validated (CV) prediction plot of orthogonal projection to latent structures-discriminant analysis (oPLS-DA) model of SML samples (light blue) and ULW samples (black). The model consisted of 5 LVs with 82.19 % and 95.41 % total captured X-block and Y-block variances, respectively. The CV accuracy, sensitivity and specificity were 100 %. (c) PCA scores plot showing the first three principal components of SML samples (light blue) and ULW samples (black). PCA was done based on 11 discriminant features selected by the genetic algorithm. Variance accounted for PC 1, 44.22 %; PC 2, 17.44 %; and PC 3, 11.92 %.

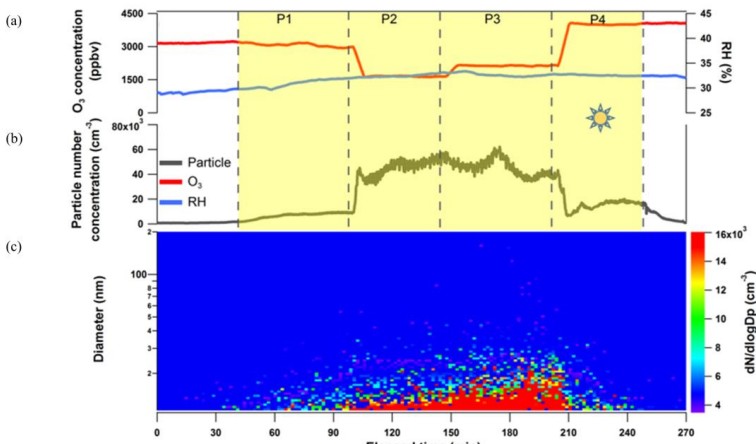

**Figure 4.** Irradiation experiment for SML CAT 8 sample in a quartz cell and subsequent particle formation from the SML interfacial gaseous products via OH radical photochemistry in the OFR. (a) O3 mixing ratio and humidity in the OFR, (b) particle concentration measured by CPC, and (c) particle size distribution profiles scanned by SMPS downstream of the OFR. The yellow shaded area represents the time period in which the quartz cell containing the concentrated SML sample was illuminated. P1-P4 corresponds to different operations to the OFR in varying oxidation degrees to the gaseous products from the quartz cell.

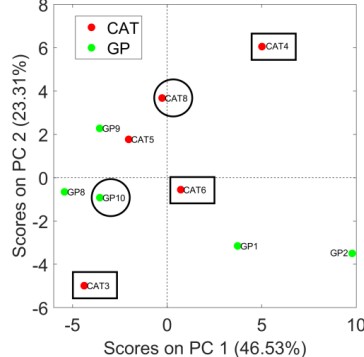

**Figure 5.** Bi-dimensional PCA score plot for SML samples using the matrix with 51 features for averaged technical replicates. Samples that were evaluated for particle formation during the Cape Verde field campaign are indicated with circles (led to SOA formation) and rectangles (did not lead to SOA formation).