# Peer review of "Seawater Analysis by Ambient Mass Spectrometry-Based Seaomics"

_Atmospheric Chemistry and Physics, 2019_

## Referee Comment (RC1) · Anonymous Referee #2 · 11 Dec 2019

This manuscript entitled "Seawater Analysis by Ambient Mass Spectrometry-Based Seaomics and Implications on Secondary Organic Aerosol Formation" by Zabalegui et al. presented a seawater "metabolomics" or "seaomics" analysis method by TM-DART-QTOF-MS. As the paper described, this method required very little sample preparation, and they were able to identify features unique to sea surface microlayer and underlying seawater. Additionally, after the untargeted chemical screening, they performed lab-to-the-field tests to look at the secondary organic aerosol potency. I appreciate they used such an experiment to add value to the untargeted chemical analysis. Based on their SOA experiment, they were able to associate certain chemical characteristics in samples with SOA formation potential. The paper presents an exciting future direction

for organic characterization for a better understanding of how organic matter can impact atmospheric processes. The paper is well written, albeit some details were lacking.

Additionally, I am concerned about:

(1) The organic matter concentrations for the TM-DART-QTOF-MS. Marine metabolomics used solid-phase extraction not only for desalting the samples but also for further concentrating the samples. Open ocean seawater required concentrating a large volume (e.g. > 1000 x concentration factor) to perform metabolomics study. With only a concentrating factor of 6.67, they may only see a very limited class of organic compounds. If the samples were collected in high productivity waters, then 6.67 might have been fine. But, without knowing the organic carbon concentration, it is hard to assess the performance of such a method. The authors should include organic carbon concentrations if available. (2) While APCI-like ionization may be less prone to salt issues, electrospray ionization covers a large range of polar compounds representing important cell metabolites. Some of such metabolites may play important roles in SOA chemistry. I would appreciate they can further comment on this so that other scientists can make informed decisions on analytical strategies for future studies. (3) While I understand 11 features were the result of aggressive feature reduction after QA/QC, this is a rather small number for "omics". Please see the specific comments below. I recommend publication of this manuscript in ACP after major revision and after the major concerns are addressed. Further specific comments are listed below.

Specific Comments: Line 40: N = 22 is rather low. The authors discussed this in the Conclusion section, which I appreciate, but probably it should be discussed earlier.

Line 42 and lines 126 to 127: 11 species are also on the low end for untargeted. See the comments below.

Lines 78 to 80: Some of these are derived from biota, consider re-structure the sentence.

Lines 176 to 177: Some of the particles need to be filtered. Centrifugation is usually not sufficient to remove all particulate matter. Please address this.

Line 178: "Extracted" may not be the correct word here. Based on the text, I assumed they meant removed. Extracted would make the reader think they have performed certain extraction protocols. Please rephrase.

Line 206 to 207: The repeat thawing and freezing process may affect the organic matter composition. What is the rationale for the thawing and re-freezing?

Lines 225 to 228: More details on sample extractions should be provided.

Line 345 and Figure 1: Based on the text, it reads like they extracted seawater. But in Figure 1, it looked like they extracted some kinds of solid (white cluster in the centrifuge tube). So, it is unclear to readers how they extract the samples.

Lines 226 and 345 to 363: Is a concentration factor of 6.67 enough? It might if the water samples were collected from high productivity water. Please include organic carbon concentration to justify this, if available.

Lines 365 to 405: The authors performed a large feature reduction, which is necessary to QA/QC untargeted data. However, from hundreds of total features down to 11 seems to be a bit aggressive. It would be nice to see how the PCA model changes at each step of feature reduction. In many untargeted environmental "omics" hundreds of features are typical even after substantial feature reduction. Therefore, it would be good to see a more detailed narrative and interpretations based on various levels of feature reduction.

---

## Referee Comment (RC2) · Anonymous Referee #1 · 2 Jan 2020

Zabalegui et al present an analysis of surface microlayer seawater samples collected from Cape Verde. Extracted SML samples were analyzed via DART-MS approaches to characterize organic species present. The intent of the work is to relate seawater organic composition (both in the SML and underlying water) to the production of VOC that can go on and form secondary organic aerosol. To this end, the investigators conducted a parallel experiment where secondary organic aerosol was measured following the OH oxidation of VOC formed from illuminated SML surfaces. The paper describes a new application of DART-MS to SML characterization and novel measurements of SML in the ocean. The paper is likely publishable in ACP, following the authors attention to the following general and specific comments.

General Comments:

The direct link between SOA formation and seawater composition is not well established chemically. A few things might help in this discussion: 1) Describe in more detail what differentiates the SML samples that lead to SOA formation (e.g., where the collected at different times, do they have different organic/inorganic ratios, surface temperature, DMS). This begs the question why a more direct experiment wasn't done to link SML composition to SOA (like measuring the VOC). 2) Figures such as F5 and S8 are confusing. Objectively, these appear to be the same figure, but yield different conclusions regarding the relationship between PC scores and SOA formation. I think a bit more discussion is needed to guide the reader through this relationship. For example, it is not clear what to take from text such as line 374: "results suggest that SML samples that led to particle formation were 375 enriched on boron-containing organic compounds and other unidentified molecules (Table S5)" In Table S5 there appears to be a slight increase in average intensity for the peaks listed, but these are likely only a very, very small subset of all the organic compounds present.

Specific Comments:

Line 53: I would suggest removing "secondary" as these processes influence the marine aerosol of both primary and secondary nature.

Line 60: The work of Bruggemann did not prove that abiotic sources of VOCs are comparable to biological sources. This was a modeling study that scaled up laboratory experiments to the global scale.

Line 91: It would be helpful to include a short discussion here on the ionization process and the bias that it can introduce when attempting a holistic analysis. DART ionization proceeds in a very similar fashion to high pressure $H_3O^+(H_2O)_n$ ion chemistry. As such, it is selective and dehydration reactions are common. It would be helpful to describe the advantages, but also the limitations when compared to ESI.

Line 135: Were surface tension measurements made to more quantitatively make this comparison? Without this (or comparable information) the dilution conditions seem arbitrary? Could it also be done with IC measurements of [Cl-]?

Line 175: What was the rationale for using negative ion mode with DART? Was positive ion mode also looked at? I was under the impression that most DART analysis was done in positive mode? Again, it would be very helpful to include some discussion of the ionization process and its selectivity.

---

## Referee Comment (RC3) · Anonymous Referee #3 · 7 Jan 2020

Synopsis:

This study describes the application of a new method (TM-DART-QTOF-MS) to the study of dissolved organic components of seawater. A major advantage of the approach is its relative insensitivity to salt, making desalination of the samples unnecessary. Data reduction techniques are used to distinguish and characterize the samples. Special attention is given to an attempt to distinguish surface microlayer (SML) composition from underlying water (ULW) composition. Finally, experiments are performed on a subset of the samples to test their ability to generate secondary organic aerosol in a photochemical reactor.

[Figure]

Broad impressions:

This manuscript is mostly easy to read, and it describes the analytical methods applied in a high level of detail. I agree with the authors that there is great potential in the presented approach, and I appreciate their thorough characterization of the analytical method.

However, not enough attention has been given to the actual biogeochemical system being studied, both in terms of the sampling methods and the interpretation. As a result, I do not think the conclusions pertaining to the SOA-forming chemistry of the surface microlayer are supported by the work presented. I think that either the scope of the work needs to be reduced and the conclusions about actual marine chemistry cut back, or the analysis and interpretation need to be expanded significantly. I recommend the former. This analytical approach shows promise; it's a proof of concept for a strategy to distinguish seawater compositional types and their potential for reactive chemistry. But this manuscript has not shown in a compelling way what actually distinguishes SML from ULW, or SOA-forming organics from non-SOA-forming organics.

Major comment 1:

The samples were frozen upon collection without any filtration. They were then thawed for analysis and centrifuged to remove large particles and colloids. This processing will result in the lysis of intact cells present in the samples, releasing dissolved compounds that will not be removed by centrifugation and will be included in the mass spectrometric analysis. One of the main reasons to perform this analysis is to understand the reactivity of the surface microlayer, and the inclusion of chemical species that were likely not available for photochemistry in the ambient environment makes this analysis difficult to interpret in that context. That is an issue that can certainly be addressed in a future study, but at the very least it needs to be discussed in this manuscript, and unless the authors can convincingly argue otherwise, it seriously limits their ability to make claims about SML reactivity. The SML can have much higher concentrations of
particles (e.g. cells) than bulk water, so the impact of this effect could be very large.

Major comment 2:

How can we infer that the handful of species that were identified as the best discriminants of SML vs. ULW are the species that are participating in important SML photochemistry and air-sea exchange? That is extremely speculative, even if the cell lysis issue discussed above is resolvable. Is there reason to think that boron-containing oxygenated organics are good SOA formers? Aren't there probably thousands of other potentially reactive compounds that covary with the SML-determining features that are identified here? There is a serious lack of discussion of these issues in this manuscript.

Minor comments:

It is not clear to this reviewer that it is necessary to coin the term "seaomics" in order to adequately describe the analysis presented.

Page 5, section 2.3: last sentence of first paragraph is hard to understand.

---

## Author Comment (AC1) · 24 Jan 2020

This manuscript entitled "Seawater Analysis by Ambient Mass Spectrometry-Based Seaomics and Implications on Secondary Organic Aerosol Formation" by Zabalegui et al. presented a seawater "metabolomics" or "seaomics" analysis method by TM-DARTQTOF-MS. As the paper described, this method required very little sample preparation, and they were able to identify features unique to sea surface microlayer and underlying seawater. Additionally, after the untargeted chemical screening, they performed lab-to-the-field tests to look at the secondary organic aerosol potency. I appreciate they used such an experiment to add value to the untargeted chemical analysis. Based on their SOA experiment, they were able to associate certain chemical characteristics in samples with SOA formation potential. The paper presents an exciting future direction for organic characterization for a better understanding of how organic matter can impact atmospheric processes. The paper is well written, albeit some details were lacking.

*We acknowledge the reviewer's feedback and helpful comments on the manuscript.*
*Just for clarification, lab-to-the-field experiments were performed during the field campaign at Cape Verde islands, whereas the untargeted chemical screening by TM-DART-QTOF-MS was performed after the campaign.*

Additionally, I am concerned about:

(1) The organic matter concentrations for the TM-DART-QTOF-MS. Marine metabolomics used solid-phase extraction not only for desalting the samples but also for further concentrating the samples. Open ocean seawater required concentrating a large volume (e.g. > 1000 x concentration factor) to perform metabolomics study. With only a concentrating factor of 6.67, they may only see a very limited class of organic compounds. If the samples were collected in high productivity waters, then 6.67 might have been fine. But, without knowing the organic carbon concentration, it is hard to assess the performance of such a method. The authors should include organic carbon concentrations if available.

*We acknowledge the suggestion provided by the reviewer. Seawater samples were collected between 500 and 1000 m away from the coastline of Bahia das Gatas. Information regarding the sampling site and dissolved organic carbon levels for SML and ULW samples are detailed in the following manuscript that provides an introduction to the MarParCloud (Marine biological production, organic aerosol Particles and marine Clouds: a process chain) campaign at the Cape Verde islands and the MARSU project, and describes the scientific content of the field campaign, the interconnection between the different facets of the project and the first findings to serve as an overview of each specific study: van Pinxteren et al., in review, 2019. As indicated in this manuscript, DOC levels varied between 1.8 and 3.2 mg $L^{-1}$ in the SML and between 0.9 and 2.8 mg $L^{-1}$ in the bulk water (Table S4 of the cited reference in review) and were in agreement with previous studies at this location (e.g. van Pinxteren et al., 2017).*

*Following the reviewer's suggestion, we will include DOC levels measured in SML and ULW samples and the cited reference at the end of section 2.2 entitled "Sample Collection at the Cape Verde Field Campaign" in the revised version of the manuscript as follows:*

"DOC levels varied between 1.8 and 3.2 mg $L^{-1}$ in the SML and between 0.9 and 2.8 mg $L^{-1}$ in the ULW water (van Pinxteren et al. in review, 2019)."

*Regarding the analytical platform used in the present study, lipophilic compounds exhibited the largest sensitivity among the different type of small molecules evaluated (Table S2). Sensitivity depends on ionization efficiency for compounds ionized with a DART source, which makes use of ionization mechanisms that predominantly follow atmospheric pressure chemical ionization (APCI)-like pathways, but in an open air format. The concentration factor selected in this study allowed organic compound extraction considering the large mass ratio between salt and organic content, and yielded 889 features (m/z) detected within samples, which were further subjected to a stringent curation process before conducting multivariate analysis. The volume of acetonitrile used for reconstituting lyophilized samples was optimized to allow enough sample volume (number of droplets and droplet volume used for depositing the sample in the mesh) that would i) allow the analysis of technical replicates, tandem MS experiments and pooled QC samples, and ii) maximize sensitivity for the maximum number of features.*

*It is worth noting that other examples in the literature for untargeted marine metabolomics have utilized a concentration factor close to 10 (e.g.: Sogin et al., 2019.).*

(2) While APCI-like ionization may be less prone to salt issues, electrospray ionization covers a large range of polar compounds representing important cell metabolites. Some of such metabolites may play important roles in SOA chemistry. I would appreciate they can further comment on this so that other scientists can make informed decisions on analytical strategies for future studies.

*As we have detailed in the introduction section:*

"It has been suggested that complex photoactive compounds are enhanced at the air-sea interface (Reeser et al., 2009a; Reeser et al., 2009b), inducing abiotic production of volatile organic compounds. For instance, experimental photosensitized reactions at the air-water interface using humic acids as a proxy of dissolved organic matter (DOM), have led to the chemical conversion of linear saturated fatty acids into unsaturated functionalized gas phase products (Ciuraru et al., 2015). Atmospheric photochemistry can even take place in the absence of photosensitizers if the air-water interface is coated with a fatty acid (Rossignol et al., 2016). On a global scale, interfacial photochemistry has recently been proven to serve as an abiotic source of volatile organic compounds comparable to marine biological emissions (Brüggemann et al., 2018)."

*Based on previous experience from the research groups involved in the present work, the analytical strategy was selected to optimize a method for lipophilic compound analysis that were proven to be involved in SOA chemistry, using a DART ionization source that is less prone than ESI to ionization suppression by high salt contents as those expected in seawater samples. This information was stated in the original version of the manuscript as follows:*

"The selected OM extraction method with acetonitrile as extracting solvent favored the analysis of lipophilic compounds. In addition, to enhance the detection of organic acids, the analytical method was optimized operating the DART ion source in negative ionization mode."
"Thermally-desorbed analytes having typically MW<1000, are ionized following atmospheric pressure chemical ionization-like pathways (Cody et al., 2005; Song et al., 2009a; Song et al., 2009b; McEwen and Larsen, 2009). An important advantage of DART compared to electrospray ionization for seawater analysis is that it is less affected by high salt levels (Kaylor et al., 2014; Tang et al., 2004), avoiding desalinization processes that may lead to sample alteration."

*The analytical strategy adopted in this work focused on minimum sample preparation and no sample desalinization, using a DART source. It is important to remark that the fraction of the marine metabolome that was covered with the implemented analytical strategy included compounds extracted in acetonitrile and subsequently ionized under APCI-like mechanisms. Regarding the selection of the ionization source, it is well known that seawater samples cannot be properly analyzed using an ESI source without a previous desalinization step. We agree with the reviewer about ESI covering a larger range of polar analytes compared to ionization sources operating under APCI-like mechanisms. Indeed, different ionization techniques are able to cover different portions of the metabolome under study. In a previous work on complex sample analysis, unique features were detected by different ionization techniques, including DART and ESI, as well as a certain degree of overlapping compounds among them (Zang et al., 2017). Accordingly, different seawater fingerprints may be obtained with different ionization techniques providing complementary information.*

*In addition, compared to a direct infusion ESI or APCI-MS-based method, in DART-MS there is no need of rinsing any tubing used to infuse liquid into the ion source. This makes DART more resistant to memory effects, minimizing carryover, as all parts in contact with the sample are disposable, and allows high-throughput analysis, as there is no need for cleaning parts between sample runs (Monge and Fernández, 2014). Another advantage of DART compared to ESI is that it mostly produces singly charged ionic species, which facilitates metabolite identification. On the other hand, ESI sources allow coupling mass spectrometry with a different orthogonal separation technique such as liquid chromatography, and hyphenated LC-MS systems provide the widest metabolome coverage with an additional dimension for compound identification, and are the most widely used analytical platforms in metabolomics (Kuehnbaum and Britz-McKibbin, 2013).*

*To further address the reviewer's comment, the following edits will be performed to the introduction section in the revised version of the manuscript (changes indicated in italics):*

"Thermally-desorbed analytes having typically MW<1000, are ionized following atmospheric pressure chemical ionization-like pathways (Cody et al., 2005; Song et al., 2009a; Song et al., 2009b; McEwen and Larsen, 2009). *Therefore, a major limitation is that it requires analytes to be volatile or semi-volatile, reducing the metabolome coverage.* An important advantage of DART compared to electrospray ionization *(ESI)* for seawater analysis is that it is less affected by high salt levels (Kaylor et al., 2014; Tang et al., 2004), avoiding desalinization processes that may lead to sample alteration. *Conversely, ESI sources allow the coupling of MS to chromatographic systems that provide an additional parameter to improve confidence in compound identification when compared to an authentic chemical standard.*"

(3) While I understand 11 features were the result of aggressive feature reduction after QA/QC, this is a rather small number for "omics".

*As we have stated in the original version of the manuscript, in the section 2.6 entitled Seaomics Data Analysis:* "Spectral features (*m/z* values) were further extracted from TM-DART-QTOF-MS data using Progenesis QI version 2.1 (Nonlinear Dynamics, Waters Corp., Milford, MA, USA). An absolute ion intensity filter was applied in the peak picking process for integration, defining a threshold for the aggregate run. Only SML and ULW samples were considered for peak picking. This process yielded 889 features (*m/z*) detected within samples."

*The subsequent curating process, using QC samples and different filtering criteria, which are also described in section 2.6 of the manuscript, yielded a 51-feature matrix. The size of the curated feature matrix exhibited a similar size as other DART-MS-based untargeted metabolomics studies focused on complex sample analysis such as aqueous samples comprised of exhaled breath condensate (e.g.: Zang et al., 2017).*

*A feature selection process was then applied to find a sub-panel of features that would allow sample classification and class membership prediction. This information was also indicated in section 2.6 of the manuscript as follows:*

"Orthogonal projection to latent structures-discriminant analysis (oPLS-DA) (Trygg et al., 2007; Bylesjö et al., 2006; Trygg and Wold, 2002; Shrestha and Vertes, 2010) coupled with a genetic algorithm (GA) variable selection method was applied to find a feature panel that maximized classification accuracy for the binary comparison of SML and ULW samples. The selected group of discriminant features had the lowest root-mean-square error of cross-validation (RMSECV) at the conclusion of the GA variable selection process. This process was performed five different times and the selected panel yielded the lowest RMSECV and exhibited largest feature overlap with the other four panels."

*In the genetic algorithm feature selection process, the maximum and minimum number of features used for isolating the discriminant panel was fixed to 15 and 5, respectively; among other parameters described in section 2.6. Metabolite identification was subsequently attempted for the 11 discriminant features resulting from the GA variable selection process.*

*The data processing, classification, prediction and analysis pipeline used in the present work is a possible strategy utilized in untargeted metabolomic studies (Clendinen et al., 2017; González-Riano et al., 2020; Broadhurst et al., 2018).*

Please see the specific comments below.
I recommend publication of this manuscript in ACP after major revision and after the major concerns are addressed. Further specific comments are listed below.

*We acknowledge the reviewer's recommendation.*

Specific Comments:

Line 40: N = 22 is rather low. The authors discussed this in the Conclusion section, which I appreciate, but probably it should be discussed earlier.

*The Conclusions section does not address limitations associated with the size of the sample cohort. The abstract of the work, however, indicates that the results obtained using a lab-to-the-field approach that were compared with those obtained using the TM-DART-QTOF-MS-based metabolomics strategy provide a proof of concept that organic compounds play a key role in aerosol formation processes at the water/air interface. In addition, the last paragraph of section 3.4 entitled "Discriminant Compound Identification & Role in Aerosol Particle Formation" addresses also the limitation associated to the low number of samples that were simultaneously analyzed by both strategies as follows:*

"Further analysis on samples analyzed by both TM-DART-QTOF-MS and the lab-to-the-field approach suggest differences in compound concentration levels between SML samples that led to SOA formation from those that did not (Fig. S8, Table S5). Figure S8A shows that PC2 clearly separates samples according to SOA formation. Those features that mainly contribute to sample class separation with largest absolute values in the loadings plot associated to PC2, and illustrated in Fig. S8B, were putatively identified as boron-containing organic compounds (Table S5). Despite the limitations associated with the low number of samples used to perform statistical analysis, results suggest that SML samples that led to particle formation were enriched on boron-containing organic compounds and other unidentified molecules (Table S5). These results provide a proof of concept that organic compounds play a key role in aerosol formation process at the water/air interface."

*Regarding the size of the sample cohort (n=22, 10 paired samples), the design of the study prioritized the analysis of collected paired samples over a larger number of non-paired samples due to variability associated with different weather conditions along the field campaign. Based on this design, ULW GP5 and ULW GP7 samples were excluded from the statistical analysis. Samples were collected during the field campaign with 2 different devices for a large number of studies that involved different analytical platforms and instrumentation both on-site at Cape Verde and after sample transportation to the different laboratories of the research groups involved in the project. We agree with the reviewer that a larger number of samples would have been desirable but the size of the sample cohort was limited by the length of the campaign, the challenges associated to sample collection, and the different types of studies that were also planned in the frame of the field campaign with these samples. More details regarding the different studies involved in the field campaign can be found in van Pinxteren, et al., in review, 2019.*

Line 42 and lines 126 to 127: 11 species are also on the low end for untargeted. See the comments below.

*The TM-DART-QTOF-MS-based untargeted metabolomics approach designed in the present study allowed extracting 889 features with unknown identity. Out of this initial matrix, 51 features were retained after the data curation process (noise filtering, LOESS correction, blank filter, CV filter in QC samples). A panel of 11 features was obtained after the feature selection process using a genetic algorithm variable selection method coupled to a cross validated oPLS-DA model, aimed at classifying and predicting samples according to their classes (SML and ULW).*

*As discussed in the Introduction section of the manuscript, targeted experiments aim to detect and quantify a predefined group of compounds with known identity. On the other hand, the untargeted metabolomics approach attempts to characterize all detectable analytes in a system, and focuses on the analysis of changes in relative abundances that generate patterns or class fingerprints without a priori knowing compound identities. The untargeted strategy utilizes multivariate statistical techniques that make use of all variables (compound features) simultaneously and deal with the relationship among them to reduce the data dimensionality, find underlying trends, and isolate those features relevant to class discrimination. Multivariate statistical methods can be supervised or unsupervised if class membership is provided or not, respectively. The chemical identification of discriminant variables contributes to the understanding of complex systems.*

*Additional details to address the reviewer's comment were already provided in the response to comment #3.*

Lines 78 to 80: Some of these are derived from biota, consider re-structure the sentence.

*We acknowledge the reviewer's remark. The statement will be modified in the revised manuscript as follows:*

"The sea surface microlayer (SML) covers up to 70 % of the Earth's surface and is enriched in DOM, including organic compounds such as fatty acids, fatty alcohols, sterols, amines, amino acids, proteins, lipids, phenolic compounds and UV-absorbing humic-like substances derived from oceanic biota; particulate matter; microorganisms (Liss and Duce, 2009; Donaldson and George, 2012); colloids and phytoplankton-exuded aggregates, mainly constituted by lipopolysaccharides, (Liss and Duce, 1997; Hunter and Liss, 1977; Bayliss and Bucat, 1975; Liss, 1986; Hardy, 1982; Garabetian et al., 1993; Williams et al., 1986; Schneider and Gagosian, 1985;Gershy, 1983; Guitart et al., 2004; Facchini et al., 2008; Kovac et al., 2002)."

Lines 176 to 177: Some of the particles need to be filtered. Centrifugation is usually not sufficient to remove all particulate matter. Please address this.

*We appreciate the reviewers' comment. In this study, the strategy was to use seawater samples with as little modification as possible. We did not intend to remove particles from the collected SML samples. Centrifugation was mainly conducted to concentrate the surface microlayer from the sea water. As expected, this step partly removed large particulate matter and colloids. But, we underline that centrifugation was aimed only at concentrating SML samples as a condition for aerosol formation. SML samples collected in the field are expected to contain different type of particles (Cunliffe et al., 2013). We agree with the reviewer that not all particles that may have been present in SML samples would have been removed with the implemented centrifugation step. The effect of particle filtration on interfacial photochemistry will be investigated in the future.*

*The revised version of the manuscript will include the following statement for clarification:*

"Centrifugation was aimed at concentrating SML samples as a condition for aerosol formation."

Line 178: "Extracted" may not be the correct word here. Based on the text, I assumed they meant removed. Extracted would make the reader think they have performed certain extraction protocols. Please rephrase.

*We agree with the reviewer's remark. The sentence will be modified in the revised manuscript as follows:*

"Subsequently, 2 mL of surface solution was collected from each centrifugal vessel to isolate closer representations of SML samples considering the dilution factor inherent to the collection process, i.e., SML diluted with ULW contribution, and leading to a total sample volume of 24 mL for subsequent experiments."

Line 206 to 207: The repeat thawing and freezing process may affect the organic matter composition. What is the rationale for the thawing and re-freezing?

*This procedure was necessary to generate sample aliquots of* *exact* *volume and pooled QC samples for further lyophilization, transportation and analysis by TM-DART-QTOF-MS at CIBION-CONICET (Argentina). As indicated in the original version of the manuscript:* "Quality control (QC) samples were prepared by mixing equal volumes of all samples including both collection methods before sample lyophilization (QC$_{ALL}$)".

*It is important to note that SML and ULW samples collected during the campaign were aliquoted on-site in bottles for the different experiments that were planned by different collaborators in the frame of the MarParCloud and the MARSU projects. Samples were stored at -20 ℃ at Cape Verde and cooled below -20 ℃ during transportation to the laboratories at TROPOS (Germany), where they were stored at -20 ℃ until they were prepared for lyophilization. A detailed description of water sampling for the different studies conducted in the frame of the campaign can be found in van Pinxteren et al., in review, 2019.*

Lines 225 to 228: More details on sample extractions should be provided.

*Following the reviewer's suggestion, more details will be included in the sample preparation description of the revised manuscript as follows:*

"Lyophilized residues were reconstituted in 1200 µL of acetonitrile, yielding a concentration factor of 6.67. Reconstituted samples were vortex-mixed during 5 min for metabolite extraction, and centrifuged during 10 min at 4861 × g and 20 °C to favor the formation of a salt pellet. For each sample, 500 µL of supernatant was collected for further analysis."

Line 345 and Figure 1: Based on the text, it reads like they extracted seawater. But in Figure 1, it looked like they extracted some kinds of solid (white cluster in the centrifuge tube). So, it is unclear to readers how they extract the samples.

*The whitish solid shown in Figure 1 illustrates the residues obtained after the lyophilization process due to the high salt content of seawater samples. Acetonitrile was the solvent selected for metabolite extraction; and a vortex-mixed step was performed to favor that process. Since a minimum amount of salt dissolves in the organic solvent, a white suspension of salt in*

*acetonitrile is formed. After centrifugation; the supernatant containing the extracted metabolites was collected without touching the salt pellet placed at the bottom of the tube, which is illustrated in the scheme of Figure 1. Supernatants were subsequently seeded on the stainless steel mesh.*

*Based on the reviewer's comment, the legend in Figure 1 will be modified in the revised version of the manuscript for clarification as follows (changes indicated in italics):*

"Scheme illustrating the analytical strategy implemented at CIBION-CONICET for the analysis of *lyophilized* seawater samples using TM-DART-QTOF-MS."

Lines 226 and 345 to 363: Is a concentration factor of 6.67 enough? It might if the water samples were collected from high productivity water. Please include organic carbon concentration to justify this, if available.

*We have already addressed this question in the response to comment #1.*

Lines 365 to 405: The authors performed a large feature reduction, which is necessary to QA/QC untargeted data. However, from hundreds of total features down to 11 seems to be a bit aggressive. It would be nice to see how the PCA model changes at each step of feature reduction. In many untargeted environmental "omics" hundreds of features are typical even after substantial feature reduction. Therefore, it would be good to see a more detailed narrative and interpretations based on various levels of feature reduction.

*A PCA score plot built with the 889 features that were initially extracted using Progenesis QI would lead to incorrect results and misunderstanding, since a large percentage of features would not follow the rigorous filters established based on signal-to-noise ratio, reproducibility and prevalence. The curation process that also included removal of signals present in blanks and signals that did not exhibit an isotopic pattern is aimed at retaining only robust signals that would increase confidence in compound annotation and subsequent data analysis in the research study (Broadhurst et al., 2018).*

*The 51-feature matrix that was retained after the curation process is comprised of the most robust features to subsequently perform multivariate statistical analysis. Figure 3A shows the score plot for the PCA model built with the 51-feature matrix obtained after the data curation process. No sample clustering was observed by using this matrix to build the model. However, the PCA model built with the 11 selected features by the genetic algorithm (Figure 3C), exhibited a certain degree of sample separation in the PC3 direction, with two sample clusters according to seawater sample collection depth, i.e., SML or ULW.*

**References**

Broadhurst, D., Goodacre, R., Reinke, S. N.,  Kuligowski, J., Wilson, I. D., Lewis, M. R., Dunn, W. B.: Guidelines and considerations for the use of system suitability and quality control samples in mass spectrometry assays applied in untargeted clinical metabolomic studies, Metabolomics 14, 72, http://dx.doi.org/10.1007/s11306-018-1367-3, 2018.

Clendinen, C. S., Monge, M. E., and Fernandez, F. M.: Ambient mass spectrometry in metabolomics, Analyst, 142, 3101-3117, http://dx.doi.org/10.1039/c7an00700k, 2017.

Cunliffe, M., Enge, A., Frka S., Gašparovic´, B.,  Guitart, C., Murrell, J. C., Salter, M. Stolle, C., Upstill-Goddard, R., Wurl, O.: Sea surface microlayers: A unified physicochemical and biological perspective of the air–ocean interface, Prog. Oceanogr., 109, 104, http://dx.doi.org/10.1016/j.pocean.2012.08.004, 2013.

González-Riano, C.,  Dudzik, D., García, A., Gil-De-La Fuente, A., Gradillas, A., Godzien, J., López-Gonzálvez, A., Rey-Stolle, F., Rojo, D., Rupérez, F. J.,  Saiz, J., and Barbas, C.,: Recent developments along the analytical process for metabolomics workflows, Anal. Chem., 92, 1, 203-226, https://doi.org/10.1021/acs.analchem.9b04553, 2020.

Kuehnbaum, N. L., Britz-McKibbin, P.: New advances in separation science for metabolomics: resolving chemical diversity in a post-genomic era, Chem. Rev. 113:2437–68, https://doi.org/10.1021/cr300484s, 2013.

Monge, M. E., and Fernández, F. M.: An Introduction to Ambient Ionization Mass Spectrometry, in: Ambient Ionization Mass Spectrometry, edited by: Domin, M. A., and Cody, R. B., The Royal Society of Chemistry, RSC Publishing, Cambridge, ISBN: 978-1-84973-926-9, 2014. Sogin, E. M, Puskás, E., Dubilier, N., Liebeke, M.: Marine metabolomics: a method for the non-targeted measurement of metabolites in seawater by gas-chromatography mass spectrometry, mSystems 4:e00638-19, https://doi.org/10.1128/mSystems.00638-19, 2019.

van Pinxteren, M., Fomba, K. W., Triesch, N., Stolle, C., Wurl, O., Bahlmann, E., Gong, X., Voigtländer, J., Wex, H., Robinson, T.-B., Barthel, S., Zeppenfeld, S., Hoffmann, E. H., Roveretto, M., Li, C., Grosselin, B., Daële, V., Senf, F., van Pinxteren, D., Manzi, M., Zabalegui, N., Frka, S., Gašparović, B., Pereira, R., Li, T., Wen, L., Li, J., Zhu, C., Chen, H., Chen, J., Fiedler, B., von Tümpling, W., Read, K. A., Punjabi, S., C. Lewis, A. C., Hopkins, J. R., Carpenter, L. J., Peeken, I., Rixen, T., Schulz-Bull, D., Monge, M. E., Mellouki, A., George, C., Stratmann, F., and Herrmann, H.: Marine organic matter in the remote environment of the Cape Verde Islands – An introduction and overview to the MarParCloud campaign, Atmos. Chem. Phys. Discuss., https://doi.org/10.5194/acp-2019-997, in review, 2019.

van Pinxteren, M., Barthel, S., Fomba, K., Müller, K., von Tümpling, W., and Herrmann, H.: The influence of environmental drivers on the enrichment of organic carbon in the sea surface microlayer and in submicron aerosol particles – measurements from the Atlantic Ocean, Elem. Sci. Anth., 5, https://doi.org/10.1525/elementa.225, 2017.

Zang, X., Pérez, J. J., Jones, C. M., Monge, M. E., McCarty, N. A., Stecenko, A. A., and Fernández, F. M.: Comparison of ambient and atmospheric pressure ion sources for cystic fibrosis exhaled breath condensate ion mobility-mass spectrometry metabolomics, J. Am. Soc. Mass Spectrom., 28(8), 1489-1496, https://doi.org/10.1007/s13361-017-1660-9, 2017.

---

## Author Comment (AC2) · 24 Jan 2020

*Responses to Reviewer #1:*

Zabalegui et al present an analysis of surface microlayer seawater samples collected from Cape Verde. Extracted SML samples were analyzed via DART-MS approaches to characterize organic species present. The intent of the work is to relate seawater organic composition (both in the SML and underlying water) to the production of VOC that can go on and form secondary organic aerosol. To this end, the investigators conducted a parallel experiment where secondary organic aerosol was measured following the OH oxidation of VOC formed from illuminated SML surfaces. The paper describes a new application of DART-MS to SML characterization and novel measurements of SML in the ocean. The paper is likely publishable in ACP, following the authors attention to the following general and specific comments.

General Comments:

The direct link between SOA formation and seawater composition is not well established chemically. A few things might help in this discussion:

1) Describe in more detail what differentiates the SML samples that lead to SOA formation (e.g., where the collected at different times, do they have different organic/inorganic ratios, surface temperature, DMS). This begs the question why a more direct experiment wasn't done to link SML composition to SOA (like measuring the VOC).

*Marine trace gases such as dimethyl sulphide (DMS), VOCs and oxygenated (O)VOCs were measured in the frame of the field campaign conducted at the Cape Verde islands and discussed in the manuscript by van Pinxteren et al. that is available online in ACPD (van Pinxteren et al., in review, 2019). This manuscript describes the scientific content of the field campaign, the interconnection between the different facets of the MarParCloud and MARSU projects and the first findings to serve as an overview of each specific study such as the one described in the present work. This overview paper will be cited in the revised version of the manuscript to complement the information about the field campaign measurements and first results.*

*Lab-to-the-field experiments conducted to explore secondary aerosol formation potency were performed on a reduced number of SML samples (n=5) that were subsequently interrogated with the TM-DART-QTOF-MS-based untargeted metabolomics analytical strategy. Out of 5 tested samples, 2 led to SOA formation. Therefore, the number of analyzed samples is low to conclude on statistically significant differences from the values obtained from the measured parameters as suggested by the reviewer (sample collection date, time and temperature were presented in Table S1; and dissolved organic carbon values can be found in van Pinxteren et al., in review, 2019). However, the results obtained in the present study from both types of experiments provide a proof of concept that organic compounds may play a key role in aerosol formation processes at the water/air interface, in agreement with previous laboratory studies that demonstrated air-sea interfacial driven chemistry as a source of marine secondary aerosol (Roveretto et al., 2019; Ciuraru et al., 2015; Fu et al., 2015).*

*All in all, we agree to the reviewer, but direct in-situ VOC measurements were not within the scope of the MARPARCLOUD field work.*

2) Figures such as F5 and S8 are confusing. Objectively, these appear to be the same figure, but yield different conclusions regarding the relationship between PC scores and SOA formation. I think a bit more discussion is needed to guide the reader through this relationship. For example, it is not clear what to take from text such as line 374: "results suggest that SML samples that led to particle formation were 375 enriched on boron-containing organic compounds and other unidentified molecules (Table S5)" In Table S5 there appears to be a slight increase in average intensity for the peaks listed, but these are likely only a very, very small subset of all the organic compounds present.

*Multivariate statistical techniques make use of all variables (compound features) simultaneously and deal with the relationship among them to reduce the data dimensionality, find underlying trends, and isolate those features relevant to class discrimination. Multivariate statistical methods can be supervised or unsupervised if class membership is provided or not, respectively.*

*Figure 5 shows a PCA scores plot of all SML samples analyzed by the TM-DART-QTOF-MS-based untargeted metabolomics strategy using the set of 51 features for averaged technical replicates. As indicated in the legend of Fig. 5, samples that were evaluated for particle formation during the Cape Verde field campaign were indicated with circles for those that led to SOA formation and rectangles for those that did not lead to SOA formation. Figure 5 shows that SML samples were not distinguished based on the collection method, i.e., GP or CAT (as it was clearly stated in line 366), and that those samples that were also evaluated for SOA formation during the field campaign and that led to aerosol formation were separated in the bidimensional scores map from those samples that did not yield to aerosol formation. Therefore, a further PCA model was built only with those samples (n=5) that were analyzed by both the lab-to-the-field approach and by TM-DART-QTOF-MS to explore sample clustering according to their feasibility of leading to particle formation using PCA, which is an unsupervised multivariate statistical method. Fig. S8A shows the PCA score plot of those 5 samples that were separated in different areas of the bidimensional map based on PC2 values according to their aerosol formation potency. Therefore, the loadings plot associated to PC2 was explored. Figures S8B showed that 7 out of 51 variables exhibited the largest weights for sample class separation based on a threshold applied to the PC2-associated loading values. Putative identification of these 7 features suggested the presence of boron-containing oxygenated organic compounds. By inspecting the relative levels of these features in both types of sample classes, an enrichment trend is observed for samples that led to SOA formation. Despite the limitation regarding the low number of samples (n=5) analyzed by both the lab-to-the field and the TM-DART-QTOF-MS approaches (discussed at the end of section 3.4 of the manuscript), and although compounds were only putatively annotated, the information reported in the work is considered to be a valuable result. Further investigation of this result is deserved in future studies considering that, as stated in the manuscript, "boron-containing compounds are known to be ubiquitous in vascular plants, marine algal species, and microorganisms (Dembitsky et al., 2002)".*

*To clarify a possible confusion suggested by the reviewer regarding PCA plots, the last paragraph of section 3.4 of the manuscript will be modified in the revised version of the manuscript as follows (inserted text shown in italics):*

"Putative identification of the discriminant panel capable of differentiating SML from ULW samples provides further evidence to support secondary organic aerosol (SOA) formation detected with the lab-to-the-field approach during the campaign. The PCA scores plot illustrated in Fig. 5 shows that SML samples were not distinguished based on the collection method, i.e., GP or CAT, and points out those SML samples that were also evaluated for SOA formation during the field campaign. As previously discussed, two of these SML samples (CAT8 and GP10) yielded SOA formation (Fig. 4). *Since CAT8 and GP10 were separated in the bidimensional scores map from the group formed of CAT3, CAT4 and CAT6; a further PCA model was built only with those samples (n=5) that were analyzed by both TM-DART-QTOF-MS and the lab-to-the-field approach (Fig. S8).* Figure S8A shows that PC2 clearly separates samples according to SOA formation. *Four out of 7 features* that mainly contribute to sample class separation with largest absolute values in the loadings plot associated to PC2, and illustrated in Fig. S8B, were putatively identified as boron-containing organic compounds (Table S5). Despite the limitations associated with the low number of samples used to perform statistical analysis, results suggest that SML samples that led to particle formation were enriched on boron-containing organic compounds and other unidentified molecules (Table S5). These results provide a proof of concept that organic compounds play a key role in aerosol formation process at the water/air interface."

Specific Comments:

Line 53: I would suggest removing "secondary" as these processes influence the marine aerosol of both primary and secondary nature.

*We appreciate the reviewer's suggestion. The word "secondary" will be removed from the statement in the revised version of the manuscript.*

Line 60: The work of Bruggemann did not prove that abiotic sources of VOCs are comparable to biological sources. This was a modeling study that scaled up laboratory experiments to the global scale.

*We agree with the reviewer's remark. The statement will be modified in the revised version of the manuscript as follows:*

"On a global scale, interfacial photochemistry has recently been suggested to serve as an abiotic source of volatile organic compounds comparable to marine biological emissions (Brüggemann et al., 2018)."

Line 91: It would be helpful to include a short discussion here on the ionization process and the bias that it can introduce when attempting a holistic analysis. DART ionization proceeds in a very similar fashion to high pressure $H_3O^+(H_2O)_n$ ion chemistry. As such, it is selective and dehydration reactions are common. It would be helpful to describe the advantages, but also the limitations when compared to ESI.

*Negative ionization DART follows negative ionization APCI-like mechanisms including electron capture, dissociative electron capture, proton abstraction, and anion adduction, which support the ionic species detected in the present study. Dehydration reactions are more commonly observed in positive ionization mode. Several*

*publications in the literature including 2 publications that were cited in the manuscript, discuss in detail DART mechanisms and the use of the negative ionization mode (McEwen and Larsen, 2009; Gross, 2014). The following additional reference will be included in the revised version of the manuscript to support the use of negative ionization DART-MS: Cody and Dane, 2013.*

*We agree with the reviewer that the ionization process introduces bias towards the fraction of the metabolome that can be interrogated with each mass spectrometry-based analytical platform. Indeed, different ionization techniques are able to cover different portions of the metabolome under study. Thus, different seawater fingerprints may be obtained with different ionization techniques providing complementary information. The strategy adopted in this work focused on minimum sample preparation and no sample desalinization by using a DART source that is less prone than ESI to ionization suppression by high salt contents as those expected in seawater samples. It is well known that seawater samples cannot be properly analyzed using an ESI source without a previous desalinization step.*

*The mechanisms operating in a DART ion source involve thermal desorption followed by plasma ionization. Therefore, a major limitation is that it requires analytes to be volatile or semi-volatile. In this regard, ESI offers the advantage of a covering more polar compounds than ionization sources operating under APCI-like mechanisms such as DART. The fraction of the marine metabolome that was covered with the implemented analytical strategy included lipophilic compounds extracted in acetonitrile and subsequently ionized under APCI-like mechanisms, limiting the analysis of more polar compounds. However, lipophilic compounds were proven to be involved in SOA chemistry and therefore consisted in an attractive fraction of the marine metabolome for interrogation.*

*Another limitation of the implemented analytical strategy is associated to compound identification and this was described in line 333 of the manuscript as follows:*

"An expected limitation of TM-DART-QTOF-MS analysis was associated to spectral overlap; thus, in some cases the isotopic pattern was not considered for compound identification."

*Not all advantages of DART-MS were included in the manuscript. Compared to a direct infusion ESI-MS- or APCI-MS-based method, in DART-MS there is no need of rinsing any tubing used to infuse liquid into the ion source. This makes DART more resistant to memory effects, minimizing carryover, as all parts in contact with the sample are disposable, and allows high-throughput analysis, as there is no need for cleaning parts between sample runs (Monge and Fernández, 2014). Another advantage of DART compared to ESI is that it mostly produces singly charged ionic species, which facilitates metabolite identification. On the other hand, ESI sources allow coupling mass spectrometry with a different orthogonal separation technique such as liquid*

*chromatography, and hyphenated LC-MS systems provide the widest metabolome coverage with an additional dimension for compound identification, and are the most widely used analytical platforms in metabolomics (Kuehnbaum and Britz-McKibbin, 2013). In this regard, LC-ESI-HRMS-based methods provide the retention time as an additional orthogonal parameter to accurate mass and fragmentation pattern that would improve confidence in compound identification when compared to an authentic chemical standard, if possible. This would allow achieving the highest confidence (level 1) in compound identification as suggested by the Metabolomics Standards Initiative (Sumner et al., 2007).*

*Based on the reviewer's suggestion, the following statements will be added in the revised version of the manuscript to include ionization processes that occur in a DART source operated in negative ion mode, and limitations of DART compared to ESI for metabolome coverage and identification:*

*i) Modifications to be made in the introduction section (changes indicated in italics):*

"Thermally-desorbed analytes having typically MW<1000, are ionized following atmospheric pressure chemical ionization-like pathways (Cody et al., 2005; Song et al., 2009a; Song et al., 2009b; McEwen and Larsen, 2009). *Therefore, a major limitation is that it requires analytes to be volatile or semi-volatile, reducing the metabolome coverage.* An important advantage of DART compared to electrospray ionization *(ESI)* for seawater analysis is that it is less affected by high salt levels (Kaylor et al., 2014; Tang et al., 2004), avoiding desalinization processes that may lead to sample alteration. *Conversely, ESI sources allow the coupling of MS to chromatographic systems that provide an additional parameter to improve confidence in compound identification when compared to an authentic chemical standard.*"

*ii) The end of section 3.1 will be modified as follows (changes indicated in italics):*

"The selected OM extraction method with acetonitrile as extracting solvent favored the analysis of lipophilic compounds. In addition, to enhance the detection of organic acids, the analytical method was optimized operating the DART ion source in negative ionization mode*, since it follows negative ionization APCI-like mechanisms including electron capture, dissociative electron capture, proton abstraction, and anion adduction (McEwen and Larsen, 2009; Cody and Dane, 2013; Gross, 2014).*"

Line 135: Were surface tension measurements made to more quantitatively make this comparison? Without this (or comparable information) the dilution conditions seem arbitrary? Could it also be done with IC measurements of [Cl-]?

*We appreciate the reviewer's comment, and do agree that such measurements would have provided valuable information. Unfortunately, these measurements suggested by the reviewer were not performed during the campaign before conducting the lab-to-the-field experiments.*
*As stated in the manuscript (line 135), sample centrifugation was conducted to isolate closer representations of the surface microlayer samples, considering the dilution factor inherent to the collection process, i.e., SML diluted with ULW contribution. Aerosol*

*particle formation was only detected after sample centrifugation and collection of 2 mL surface solution. That is, centrifugation was aimed at concentrating SML samples as a condition for aerosol formation.*

*The revised version of the manuscript will include the following statement to address this point:*

"Centrifugation was aimed at concentrating SML samples as a condition for aerosol formation."

Line 175: What was the rationale for using negative ion mode with DART? Was positive ion mode also looked at? I was under the impression that most DART analysis was done in positive mode? Again, it would be very helpful to include some discussion of the ionization process and its selectivity.

*As stated above, negative ionization DART follows negative ionization APCI-like mechanisms including electron capture, dissociative electron capture, proton abstraction, and anion adduction. Examples that can be found in the literature that discuss in detail DART mechanisms and the use of the negative ionization mode include: McEwen and, Larsen, 2009; Cody and Dane, 2013; Gross, 2014. Two of these papers were cited in the manuscript and the publication by Cody and Dane, 2013 will be included in the revised version of the manuscript.*

*Negative ion mode was chosen to operate the DART ion source in order to detect lipophilic compounds, which were proven to be involved in SOA chemistry. Therefore, they consisted in an attractive fraction of the marine metabolome for interrogation and testing as discriminant compounds based on their relative levels between ULW and SML samples. During the analytical method development, both ionization modes were tested with different selectivity and sensitivity for the different type of compounds analyzed (amino acids, sugars and lipids indicated in Table S2); negative ion mode being more sensitive to the analysis of lipophilic compounds, including organic acids.*

*Regarding the inclusion of discussion on the ionization process, the manuscript already included the following statements:*

Line 91: "Thermally-desorbed analytes having typically MW<1000, are ionized following atmospheric pressure chemical ionization-like pathways (Cody et al., 2005; Song et al., 2009a; Song et al., 2009b; McEwen and Larsen, 2009)."

Line 261: "In addition, to enhance the detection of organic acids, the analytical method was optimized operating the DART ion source in negative ionization mode."

Line 340: "Different types of species were generated for desorbed and ionized analytes (M) by the plasma-based source operated in negative mode, including $[M-H]^-$, $[M]^-$ and $[M]^{\bullet-}$ ionic species."

Line 345: "literature evidence suggests the production of radical anions based on electron capture mechanisms occurring in He-based plasma sources (Cody and Dane, 2016; Bridoux and Machuron-Mandard, 2013; Jorabchi et al., 2013)."

*Despite the inclusion of the previous statements described with citations in the manuscript regarding the ionization mechanisms that take place in a DART ion source operated in negative mode and following the reviewer's suggestion, the end of section 3.1 will be modified in the revised manuscript as follows (indicated in italics):*

"The selected OM extraction method with acetonitrile as extracting solvent favored the analysis of lipophilic compounds. In addition, to enhance the detection of organic acids, the analytical method was optimized operating the DART ion source in negative ionization mode*, since it follows negative ionization APCI-like mechanisms including electron capture, dissociative electron capture, proton abstraction, and anion adduction (McEwen and Larsen, 2009; Cody and Dane, 2013; Gross, 2014)."*

**References**

Ciuraru, R., Fine, L., van Pinxteren, M., D'Anna, B., Herrmann, H., and George, C.: Photosensitized production of functionalized and unsaturated organic compounds at the air-sea interface, Sci. Rep., 5, 12741, http://dx.doi.org/10.1038/srep12741, 2015.

Cody, R. B, Dane, A. J.: Soft Ionization of Saturated Hydrocarbons, Alcohols and Nonpolar Compounds by Negative-Ion Direct Analysis in Real-Time Mass Spectrometry, J. Am. Soc. Mass Spectrom. 24:329Y334, http://dx.doi.org/10.1007/s13361-012-0569-6, 2013.

Dembitsky, V. M., Smoum, R., Al-Quntar, A. A., Ali, H. A., Pergament, I., and Srebnik, M.: Natural occurrence of boron-containing compounds in plants, algae and microorganisms, Plant Science, 163, 931-942, http://dx.doi.org/10.1016/S0168-9452(02)00174-7, 2002.

Fu, H., Ciuraru, R., Dupart, Y., Passananti, M., Tinel, L., Rossignol, S., Perrier, S., Donaldson, D. J., Chen, J., and George, C.: Photosensitized Production of Atmospherically Reactive Organic Compounds at the Air/Aqueous Interface, J. Am. Chem. Soc., 137, 8348-8351, http://dx.doi.org/10.1021/jacs.5b04051, 2015.

Gross J. H.: Direct analysis in real time—a critical review on DART-MS, Anal. Bioanal. Chem. 406, 63–80, http://dx.doi.org/10.1007/s00216-013-7316-0, 2014.

Kuehnbaum, N. L., Britz-McKibbin, P.: New advances in separation science for metabolomics: resolving chemical diversity in a post-genomic era, Chem. Rev. 113:2437–68, https://doi.org/10.1021/cr300484s, 2013.

McEwen, C. N., Larsen, B. S.: Ionization mechanisms related to negative ion APPI, APCI, and DART. J. Am. Soc. Mass Spectrom. 20, 1518–1521, http://dx.doi.org/10.1016/j.jasms.2009.04.010, 2009.

Monge, M. E., and Fernández, F. M.: An Introduction to Ambient Ionization Mass Spectrometry, in: Ambient Ionization Mass Spectrometry, edited by: Domin, M. A., and Cody, R. B., The Royal Society of Chemistry, RSC Publishing, Cambridge, ISBN: 978-1-84973-926-9, 2014.

Roveretto, M., Li, M., Hayeck, N., Brüggemann, M., Emmelin, C., Perrier, S., and George, C.: Real-Time Detection of Gas-Phase Organohalogens from Aqueous Photochemistry Using Orbitrap Mass Spectrometry, ACS Earth and Space Chemistry, 3, 329-334, http://dx.doi.org/10.1021/acsearthspacechem.8b00209, 2019.

Sumner, L.W., et al: Proposed minimum reporting standards for chemical analysis Chemical Analysis Working Group (CAWG) Metabolomics Standards Initiative (MSI), Metabolomics, 3(3), 211-221, http://dx.doi.org/10.1007/s11306-007-0082-2, 2007.

van Pinxteren, M., Fomba, K. W., Triesch, N., Stolle, C., Wurl, O., Bahlmann, E., Gong, X., Voigtländer, J., Wex, H., Robinson, T.-B., Barthel, S., Zeppenfeld, S., Hoffmann, E. H., Roveretto, M., Li, C., Grosselin, B., Daële, V., Senf, F., van Pinxteren, D., Manzi, M., Zabalegui, N., Frka, S., Gašparović, B., Pereira, R., Li, T., Wen, L., Li, J., Zhu, C., Chen, H., Chen, J., Fiedler, B., von Tümpling, W., Read, K. A., Punjabi, S., C. Lewis, A. C., Hopkins, J. R., Carpenter, L. J., Peeken, I., Rixen, T., Schulz-Bull, D., Monge, M. E., Mellouki, A., George, C., Stratmann, F., and Herrmann, H.: Marine organic matter in the remote environment of the Cape Verde Islands – An introduction and overview to the MarParCloud campaign, Atmos. Chem. Phys. Discuss., https://doi.org/10.5194/acp-2019-997, in review, 2019.

---

## Author Comment (AC3) · 24 Jan 2020

*Responses to Reviewer #3:*

Synopsis:

This study describes the application of a new method (TM-DART-QTOF-MS) to the study of dissolved organic components of seawater. A major advantage of the approach is its relative insensitivity to salt, making desalination of the samples unnecessary. Data reduction techniques are used to distinguish and characterize the samples. Special attention is given to an attempt to distinguish surface microlayer (SML) composition from underlying water (ULW) composition. Finally, experiments are performed on a subset of the samples to test their ability to generate secondary organic aerosol in a photochemical reactor.

*Just for clarification, lab-to-the-field experiments were performed during the field campaign at Cape Verde islands, whereas the untargeted chemical screening by TM-DART-QTOF-MS was performed after the campaign.*

Broad impressions:

This manuscript is mostly easy to read, and it describes the analytical methods applied in a high level of detail. I agree with the authors that there is great potential in the presented approach, and I appreciate their thorough characterization of the analytical method. However, not enough attention has been given to the actual biogeochemical system being studied, both in terms of the sampling methods and the interpretation. As a result, I do not think the conclusions pertaining to the SOA-forming chemistry of the surface microlayer are supported by the work presented. I think that either the scope of the work needs to be reduced and the conclusions about actual marine chemistry cut back, or the analysis and interpretation need to be expanded significantly. I recommend the former. This analytical approach shows promise; it's a proof of concept for a strategy to distinguish seawater compositional types and their potential for reactive chemistry. But this manuscript has not shown in a compelling way what actually distinguishes SML from ULW, or SOA-forming organics from non-SOA-forming organics.

*We acknowledge the reviewer's observations to the work described in the submitted manuscript. To clarify some aspects of the comments provided by the reviewer, we describe the objectives of the work:*
*i) to develop an ambient mass spectrometry-based untargeted metabolomics method that would allow a comprehensive screening of seawater samples with no need of desalination using a DART ionization source operated in negative mode and coupled to a high resolution mass spectrometer;*
*ii) to isolate by means of multivariate statistical methods a panel of ionic species that were present in both SML and ULW samples but based on their relative levels they differentiated seawater samples according to their collection depth (i.e., SML or ULW);*
*iii) to provide putative identification of these discriminant ionic species (based on the ionic species detected according to the ionization mechanisms operating in a negative mode DART ion source, based on accurate mass values and on isotopic patterns) to understand their possible implication in secondary organic aerosol (SOA) formation*

*processes at the water/air interface based on their functional groups and chemical families;*
*iv) to develop a lab-to-the-field approach to evaluate the SOA formation potency of SML samples;*
*v) to apply multivariate statistical methods to analyze the data acquired by TM-DART-QTOF-MS from the subset of SML samples that were also analyzed in-site by a lab-to-the-field approach;*
*vi) to isolate by means of Principal Component Analysis, which is a non-supervised method, those features (ionic species) with largest weight in differentiating samples that lead to particle formation from those that did not lead to particle formation according to the results from field experiments;*
*vii) to provide putative identification to those discriminant features;*
*viii) to connect the results obtained from both type of experiments, i.e., the seaomics and the lab-to-the-field approaches, which consist in two different and complementary strategies.*

*As clearly stated in the whole manuscript and summarized in the abstract, a panel of 11 ionic species detected in all seawater samples (SML and ULW) allowed sample class discrimination by means of supervised multivariate statistical models. Tentative identification of these species suggest that saturated fatty acids, peptides, fatty alcohols, halogenated compounds, and oxygenated boron-containing organic compounds may be involved in water-air transfer processes and in photochemical reactions at the water-air interface of the ocean. Results from the lab-to-the-field experiments conducted to explore secondary aerosol formation potency of a reduced number of samples (n=5), which were also subsequently interrogated with the TM-DART-QTOF-MS-based untargeted metabolomics analytical strategy, provide a proof of concept that organic compounds may play a key role in aerosol formation processes at the water/air interface. We consider that these results do contribute to the chemical characterization of the sea surface microlayer composition through the implementation of a developed mass spectrometry-based untargeted metabolomics analytical method utilizing an ambient ionization source. We also consider that experiments conducted in-site to evaluate SOA formation potency of SML samples add value to the untargeted chemical analysis. Therefore, seaomics results were discussed in terms of their implications on SOA formation. To our understanding, the novel analytical method and the results described in the manuscript may be interesting for the scientific community and should therefore be considered for publication in ACP.*

*More studies have been and are planned to be conducted on the same seawater samples for a deeper and complementary characterization of SML and ULW samples, and to understand to what extent is seawater a source of marine organic matter on aerosol particles and cloud water, as described in the following overview paper: van Pinxteren et al., in review, 2019. This paper provides an introduction to the MarParCloud (Marine biological production, organic aerosol Particles and marine Clouds: a process chain) campaign at the Cape Verde islands and the MARSU project, describing the scientific content of the field campaign, the interconnection between the different facets of the project and the first findings to serve as an overview of each specific study. A detailed description of water sampling methods for the different studies conducted in the frame of the campaign can be found in the cited manuscript. This paper will be cited in the revised version of the manuscript.*

*In line with the previous discussion, and to address the reviewer's suggestions, the following changes will be done on the revised manuscript (edits indicated in italics):*

*i) Abstract:*

"Results from these experiments and the analytical seaomics strategy provide a proof of concept that organic compounds *may* play a key role in aerosol formation processes at the water/air interface."

*ii) Section 3.4, Line 375:*

"These results provide a proof of concept that organic compounds *may* play a key role in aerosol formation process at the water/air interface."

*iii) Conclusions:*

"Combined results from TM-DART-QTOF-MS and on-site SOA formation testing experiments on SML samples, suggest that organic compounds enriched at the water/air interface *may* play a key role in aerosol formation process."

Major comment 1:

The samples were frozen upon collection without any filtration. They were then thawed for analysis and centrifuged to remove large particles and colloids. This processing will result in the lysis of intact cells present in the samples, releasing dissolved compounds that will not be removed by centrifugation and will be included in the mass spectrometric analysis. One of the main reasons to perform this analysis is to understand the reactivity of the surface microlayer, and the inclusion of chemical species that were likely not available for photochemistry in the ambient environment makes this analysis difficult to interpret in that context. That is an issue that can certainly be addressed in a future study, but at the very least it needs to be discussed in this manuscript, and unless the authors can convincingly argue otherwise, it seriously limits their ability to make claims about SML reactivity. The SML can have much higher concentrations of particles (e.g. cells) than bulk water, so the impact of this effect could be very large.

*We agree with the reviewer that the fact that samples were not filtered immediately after collection suggest that the analytical method was utilized for a comprehensive analysis of the marine metabolome including both the endo and exometabolome. Actually, this information was stated in line 361 of the manuscript as follows:*

"It is worth noting that organic compounds identified in the discriminant panel may have derived both from the secreted (exometabolome) and/or intracellular metabolites (endometabolome) of biological organisms such as algal species and microorganisms present in seawater."

*Since all samples were identically treated, the results are useful to illustrate relative differences in metabolites that are present in all samples collected at different depths (i.e., ULW and SML). As indicated in Table S4, fold changes were calculated for*

*discriminant features detected in all samples (both SML and ULW). Within the 11 discriminant features detected in all samples, 3 were statistically enriched in SML samples and 2 were statistically enriched in ULW samples with 7 out of 11 exhibiting positive fold changes towards SML samples.*

*The manuscript clearly indicates that results obtained with the seaomics and the lab-to-the-field approaches provide a proof of concept that organic compounds may play a key role in aerosol formation process at the water/air interface. In addition, results from the atmospheric simulation experiments conducted on SML samples were in agreement with previous laboratory studies that demonstrated air-sea interfacial driven chemistry as a source of marine secondary aerosol (Roveretto et al., 2019; Ciuraru et al., 2015; Fu et al., 2015).*

*In agreement with the reviewer's remark, a different study may be conducted in the future with a different aim and design, based on the results obtained in the present study.*

*Further analyses of these seawater samples have been and will be conducted with additional analytical platforms to provide a complementary characterization of SML and ULW samples as detailed in van Pinxteren et al., in review, 2019. For example, enrichment factors obtained for bacterial abundance in SML samples ranged between 0.88 and 1.21 (van Pinxteren et al., in review, 2019) despite the expected larger concentrations, as suggested by the reviewer.*

*Regarding the experiments performed in-site during the field campaign with the lab-to-the-field approach to evaluate the feasibility of SML samples to lead to SOA formation, the strategy was to use seawater samples with as little modification as possible. Centrifugation was mainly conducted to concentrate the surface microlayer from the seawater. As expected, this step partly removed large particulate matter and colloids. But, centrifugation was aimed only at concentrating SML samples as a condition for aerosol formation.*

*The revised version of the manuscript will include the following statement for clarification:*

"Centrifugation was aimed at concentrating SML samples as a condition for aerosol formation."

Major comment 2:

How can we infer that the handful of species that were identified as the best discriminants of SML vs. ULW are the species that are participating in important SML photochemistry and air-sea exchange? That is extremely speculative, even if the cell lysis issue discussed above is resolvable. Is there reason to think that boron-containing oxygenated organics are good SOA formers? Aren't there probably thousands of other potentially reactive compounds that covary with the SML-determining features that are identified here? There is a serious lack of discussion of these issues in this manuscript.

*The best discriminant feature panel for SML vs. ULW samples comprises species that may participate in SML photochemistry and air-sea exchange based on the functional*

*groups provided by the putatively identified ionic species in agreement with previous evidence reported in the literature cited in the manuscript. The following statements addressed in the original version of the manuscript the potential role of putatively identified discriminant compounds in SOA formation processes:*

Line 38: "A panel of 11 ionic species detected in all samples allowed sample class discrimination by means of supervised multivariate statistical models. Tentative identification of these species suggests that saturated fatty acids, peptides, fatty alcohols, halogenated compounds, and oxygenated boron-containing organic compounds may be involved in water-air transfer processes and in photochemical reactions at the water-air interface of the ocean."

Line 359: "Possible sources of halogenated compounds in SML samples are photochemical reactions occurring at the water/air interface (Roveretto et al., 2019; Donaldson and George, 2012)."

Line 364: "Putative identification of the discriminant panel capable of differentiating SML from ULW samples provides further evidence to support secondary organic aerosol (SOA) formation detected with the lab-to-the-field approach during the campaign."

Line 381: "Tentative identification of the discriminant metabolite panel suggests that halogenated compounds, fatty alcohols, and oxygenated boron-containing organic compounds may be involved in water-air transfer processes and in photochemical reactions at the water-air interface of the ocean."

*We also clearly stated at the end of section 3.4 (Discriminant Compound Identification & Role in Aerosol Particle Formation) the limitation associated to the low number of samples (n=5) that were able to be analyzed by both the lab-to-the field approach and the TM-DART-QTOF-MS approach. We still consider an interesting result the fact that by implementing a non-supervised multivariate statistical method such as PCA, samples were able to be grouped according to their feasibility of leading to particle formation as identified with the lab-to-the-field approach. The manuscript does not claim that boron-containing oxygenated organics are good SOA formers. The putative identification of features with the largest weight in the loadings plot associated to PC2 in Figure S8 suggest the presence of boron-containing oxygenated organic compounds. The manuscript clearly indicates in Table S5 that boron-containing oxygenated organic compounds are 4 putatively identified compounds based on accurate mass and isotopic pattern analysis out of 7 discriminat features selected based on a threshold applied to the PC2-associated loading values. These compounds were only putatively annotated but still the information is considered to be a valuable result that deserves further investigation in future studies since, as stated in the manuscript, "boron-containing compounds are known to be ubiquitous in vascular plants, marine algal species, and microorganisms (Dembitsky et al., 2002)".*

Minor comments:

It is not clear to this reviewer that it is necessary to coin the term "seaomics" in order to adequately describe the analysis presented.

*We believe that "seaomics" is a simplified but straightforward term that summarizes the untargeted metabolomics strategy utilized to interrogate seawater samples.*

Page 5, section 2.3: last sentence of first paragraph is hard to understand.

*We agree with the reviewer's remark. The sentence will be modified in the revised manuscript as follows:*

"Subsequently, 2 mL of surface solution was collected from each centrifugal vessel to isolate closer representations of SML samples considering the dilution factor inherent to the collection process, i.e., SML diluted with ULW contribution, and leading to a total sample volume of 24 mL for subsequent experiments."

**References**

Ciuraru, R., Fine, L., van Pinxteren, M., D'Anna, B., Herrmann, H., and George, C.: Photosensitized production of functionalized and unsaturated organic compounds at the air-sea interface, Sci. Rep., 5, 12741, http://dx.doi.org/10.1038/srep12741, 2015.

Dembitsky, V. M., Smoum, R., Al-Quntar, A. A., Ali, H. A., Pergament, I., and Srebnik, M.: Natural occurrence of boron-containing compounds in plants, algae and microorganisms, Plant Science, 163, 931-942, http://dx.doi.org/10.1016/S0168-9452(02)00174-7, 2002.

Fu, H., Ciuraru, R., Dupart, Y., Passananti, M., Tinel, L., Rossignol, S., Perrier, S., Donaldson, D. J., Chen, J., and George, C.: Photosensitized Production of Atmospherically Reactive Organic Compounds at the Air/Aqueous Interface, J. Am. Chem. Soc., 137, 8348-8351, http://dx.doi.org/10.1021/jacs.5b04051, 2015.

Roveretto, M., Li, M., Hayeck, N., Brüggemann, M., Emmelin, C., Perrier, S., and George, C.: Real-Time Detection of Gas-Phase Organohalogens from Aqueous Photochemistry Using Orbitrap Mass Spectrometry, ACS Earth and Space Chemistry, 3, 329-334, http://dx.doi.org/10.1021/acsearthspacechem.8b00209, 2019.

van Pinxteren, M., Fomba, K. W., Triesch, N., Stolle, C., Wurl, O., Bahlmann, E., Gong, X., Voigtländer, J., Wex, H., Robinson, T.-B., Barthel, S., Zeppenfeld, S., Hoffmann, E. H., Roveretto, M., Li, C., Grosselin, B., Daële, V., Senf, F., van Pinxteren, D., Manzi, M., Zabalegui, N., Frka, S., Gašparović, B., Pereira, R., Li, T., Wen, L., Li, J., Zhu, C., Chen, H., Chen, J., Fiedler, B., von Tümpling, W., Read, K. A., Punjabi, S., C. Lewis, A. C., Hopkins, J. R., Carpenter, L. J., Peeken, I., Rixen, T., Schulz-Bull, D., Monge, M. E., Mellouki, A., George, C., Stratmann, F., and Herrmann, H.: Marine organic matter in the remote environment of the Cape Verde Islands – An

introduction and overview to the MarParCloud campaign, Atmos. Chem. Phys. Discuss., https://doi.org/10.5194/acp-2019-997, in review, 2019.

---

## Author Response (AR2)

April 2nd, 2020
Ciudad de Buenos Aires, Argentina

Dr. P. Zieger
Editor, Atmospheric Chemistry and Physics

Dear Editor,

I am pleased to submit the revised version of our manuscript entitled "Seawater Analysis by Ambient Mass Spectrometry-Based Seaomics" for its publication as an article in the Atmospheric Chemistry and Physics Special Issue entitled "Marine organic matter: from biological production in the ocean to organic aerosol particles and marine clouds" (ACP/OS inter-journal SI).

We acknowledge the reviewer's comments as well as your suggestions regarding the changes that still needed to be made in the manuscript to be considered for publication in ACP. Accordingly, we have thoroughly followed all recommendations. We have changed the title as suggested and addressed all comments and suggestions made by reviewer #3 to properly tone down the link between the seawater composition measurements and SOA formation.

A highlighted version of the manuscript has been prepared using track changes. Please, find at the bottom of this letter a point-by-point response to the reviewer's comments and the marked-up manuscript version showing the changes made.

We believe that this work can be of interest to the atmospheric chemistry community, since we present a new approach to bridge different fields of science studying the air/sea interface.

I hope that this modified version of the manuscript can be finally accepted for publication in ACP.

Sincerely,

María Eugenia Monge, Ph.D.

Principal Investigator, Research Staff CONICET
Centro de Investigaciones en Bionanociencias
Godoy Cruz 2390, C1425FQD CABA, Argentina
Phone: +54 11 4899-5500 ext 5614
E-mail: maria.monge@cibion.conicet.gov.ar

This manuscript looks almost identical to the one I reviewed previously, and the main strengths and weaknesses largely remain. It is a highly descriptive account of a novel method of characterizing sea surface microlayer (SML) and bulk seawater composition with the aim of ultimately connecting surface ocean composition with marine volatile organic compound and aerosol formation. Attached to this, with limited analysis and synthesis (but large weighting in the study conclusions), is an experiment that tested secondary organic aerosol (SOA) generation by irradiation of a subset of the SML samples.

The authors seem not to have taken to heart the criticism of the reviewers that the connections between the composition measurements and the SOA formation experiments are very tenuous. Put simply, there is nothing that shows that the detected compounds enhanced in the SML are contributing to SOA formation. I proposed in my original review that the authors either dramatically expand the discussion/analysis, or temper their conclusions. The authors seem not to have been persuaded that there was much of a need for either. They have held back from meaningfully editing the manuscript to make their arguments clearer or more convincing to the reader.

I see three possible paths to publication for this manuscript:

1. Change the conclusions in the Conclusions and Abstract to things clearly supported by the manuscript without other major changes (suggestions below). It's not clear if the manuscript in that form would be suitable for publication in ACP or not, as little could be concluded about the relationship between the seawater composition measurements and SOA formation.

2. Dramatically expand the analysis and interpretation and synthesis of the SOA-related section to draw a compelling link to the "seaomics" work. This may or may not be doable given the available data, but it could lead to material support for the conclusions as currently stated.

3. Remove the SOA-focused experiments from the paper and submit the detailed, useful description of the SML and bulk seawater composition method and results to a more appropriate journal.

*We acknowledge the reviewer's comments and constructive suggestions regarding the changes that still needed to be made in the manuscript. We understand the concerns pointed out by the reviewer and have accordingly followed all recommendations thoroughly and toned down the link between the seawater composition measurements and SOA formation experiments. We have changed the title of the manuscript to "Seawater Analysis by Ambient Mass Spectrometry-Based Seaomics", and we have tempered the conclusions in the Conclusions and Abstract sections.*

*We believe that the revised version of the manuscript will be of interest to the atmospheric chemistry community, since this work presents a new approach to bridge different fields of science studying the air/sea interface.*

Specific major Issues with claims in the manuscript by section:

Abstract:
"Tentative identification of these species suggest that saturated fatty acids, peptides, fatty alcohols, halogenated compounds, and oxygenated boron-containing organic compounds may be involved in water-air transfer processes and in photochemical reactions at the water- air interface of the ocean."

I would revise to "…boron-containing organic compounds are available at the surface for water-air transfer processes…" There's no evidence here that these things are specifically participating- they are simply present.

"Results from these experiments and the analytical seaomics strategy provide a proof of concept that organic compounds may play a key role in aerosol formation processes at the water/air interface."

I would revise to "These results provide a proof on concept for an approach to identifying organic molecules involved in aerosol formation processes at the water/air interface." Writing "proof of concept" isn't a license to state a desired conclusion that isn't supported by the present data.

*We have thoroughly modified the abstract following the reviewer's advice as follows (changes indicated in italics):*

*"A transmission mode-direct analysis in real time-quadrupole time of flight-mass spectrometry (TM-DART-QTOF-MS)-based analytical method coupled to multivariate statistical analysis was developed to interrogate lipophilic compounds in seawater samples without the need of desalinization. An untargeted metabolomics approach addressed here as seaomics was successfully implemented to discriminate sea surface*

microlayer (SML) from underlying water (ULW) samples (n=22, 10 paired samples) collected during a field campaign at the Cape Verde islands in September-October 2017. A panel of 11 ionic species detected in all samples allowed sample class discrimination by means of supervised multivariate statistical models. Tentative identification of species *enriched at SML samples* suggests that fatty alcohols, halogenated compounds, and oxygenated boron-containing organic compounds *are available at the surface for water-air transfer processes*. A subset of SML samples (n=5) were subject to on-site experiments during the campaign using a lab-to-the-field approach to test their secondary organic aerosol (SOA) formation potency. Results from these experiments and the analytical seaomics strategy provide a proof of concept *for an approach to identifying organic molecules involved in aerosol formation processes at the water/air interface.*

Line 425:

I am repeating myself from my previous review, but the compounds present in SML particles ("endometabolome") may not be available to participate in the sea surface SOA-forming chemistry in the real environment. That should be pointed out to the reader with at least a single sentence. It is highly relevant to the question of how relevant these results may be to the ambient environment, and glossing over it (i.e. merely pointing out in review that those components are included in the analysis) does a disservice to the reader.

*We have clarified the point raised by the reviewer as follows:*

"It is worth noting that organic compounds identified in the discriminant panel may have derived both from the secreted (exometabolome) and/or intracellular metabolites (endometabolome) of biological organisms such as algal species and microorganisms present in seawater, *since samples were not filtered. Therefore, in a real environment some of these compounds may be present in lower levels than those detected in the present work or may not be available to participate in the sea surface SOA-chemistry.*"

Conclusions:
I disagree with the conclusion that the results here "suggest that organic compounds enriched at the water/air interface may play a key role in aerosol formation processes." It seems likely to be true based on previous work, but I don't see how these results support that claim. If the experiment had included OFR oxidation runs using the underlying water and showed that those samples showed no SOA formation, such a

conclusion could be supported. Lacking such data, the authors would need to be able to say something about which compounds may be contributing to the differential SOA forming ability of the different SML samples, and then argue that those compounds are enhanced in the SML.

*Following the reviewer's advice, we have modified the Conclusions section as follows:*

[revised manuscript text omitted]